# Tumor suppressor p53 regulates intestinal type 2 immunity

Chun-Yuan Chang [1,5], Jianming Wang [1,5], Yuhan Zhao [1,5], Juan Liu[1], Xue Yang[1], Xuetian Yue [1], Huaying Wang[1], Fan Zhou[1], Juan M. Inclan-Rico[2], John J. Ponessa [2], Ping Xie [3], Lanjing Zhang [1,4], Mark C. Siracusa[2], Zhaohui Feng [1✉] & Wenwei Hu [1✉]

The role of p53 in tumor suppression has been extensively studied and well-established. However, the role of p53 in parasitic infections and the intestinal type 2 immunity is unclear. Here, we report that p53 is crucial for intestinal type 2 immunity in response to the infection of parasites, such as *Tritrichomonas muris* and *Nippostrongylus brasiliensis*. Mechanistically, p53 plays a critical role in the activation of the tuft cell-IL-25-type 2 innate lymphoid cell circuit, partly via transcriptional regulation of Lrmp in tuft cells. Lrmp modulates $Ca^{2+}$ influx and IL-25 release, which are critical triggers of type 2 innate lymphoid cell response. Our results thus reveal a previously unrecognized function of p53 in regulating intestinal type 2 immunity to protect against parasitic infections, highlighting the role of p53 as a guardian of immune integrity.

[1] Rutgers Cancer Institute of New Jersey, Rutgers University, New Brunswick, NJ, USA. [2] Department of Medicine, Rutgers New Jersey Medical School, Rutgers University, Newark, NJ, USA. [3] Department of Cell Biology and Neuroscience, Rutgers University, Piscataway, NJ, USA. [4] Department of Pathology, Penn Medicine Princeton Medical Center, Plainsboro, NJ, USA. [5] These authors contributed equally: Chun-Yuan Chang, Jianming Wang, Yuhan Zhao. ✉email: fengzh@cinj.rutgers.edu; wh221@cinj.rutgers.edu

As "the guardian of the genome", the role of p53 in tumor suppression has been well-established[1,2]. p53 is the most frequently mutated gene in human tumors. As a transcriptional factor, p53 responds to various endogenous and exogenous stimuli by selectively inducing its target genes in a tissue and stress type-specific manner to regulate many important biological processes, including cell cycle arrest, apoptosis, senescence, etc., which in turn maintain genomic stability to prevent cancer[3,4]. While the key role of p53 in tumor suppression has been well-established, recent studies have revealed that p53 regulates additional important biological processes, such as fertility, stemness, metabolism, etc.[2].

p53 was originally identified as a cellular protein that interacts with the large T antigen of Simian Virus 40 (SV40), suggesting a role of p53 in the anti-viral response[5,6]. Further studies have reported that p53 is activated by viral infections, including Epstein-Barr virus, adenovirus, and HIV-1, which in turn induces apoptosis to limit viral replication and eliminate infected cells[7]. These findings support the role of p53 in anti-viral innate immunity. In addition, several studies have indicated a potential role of p53 in parasitic infections[8–12]. Plasmodium parasites cause malaria in humans and mice. A study of children with Plasmodium falciparum (Pf) infection revealed that a pre-infection signature including p53 activation. Activation of p53 is associated with control of malarial fever and parasitemia after Pf infection, and further, the enhanced p53 expression in monocytes attenuates Pf-induced inflammation[8]. Another study indicated that suppression of host p53 is critical for liver-stage infection of Plasmodium in mice; mice with increased p53 levels showed reduced liver-stage parasite burden, whereas p53 knockout mice displayed increased liver-stage burden[9]. The parasitic flatworms of Schistosoma cause schistosomiasis. Schistosoma japonicum soluble egg antigens have been reported to activate p53 in hepatic stellate cells to promote their apoptosis or senescence[10,11]. In addition, Leishmania major, a protozoan Leishmainia parasite, has been reported to increase p53 expression and induce apoptosis of host lymphocytes[12]. While studies have suggested the role of p53 in immunology (for review, see[13]), its precise function and underlying mechanisms are far from clear. It is also unclear whether p53 regulates the type 2 immune response in the context of parasitic infections.

Parasitic infections affect an estimated two billion people worldwide, which is a significant public health concern[14]. Upon the parasitic infections, tuft cells, a rare cell population (<1%) in the intestine, sense parasites and secret the cytokine IL-25. IL-25 functions as a major activating signal to recruit and induce the expansion of type 2 innate lymphoid cells (ILC2s) and type 2 helper T cells in the intestinal lamina propria (LP) to secret type 2 cytokines, including IL-4, IL-5, and IL-13, to drive the type 2 immune response[15–17]. In turn, IL-13 stimulates the differentiation of intestinal stem cells into tuft cells and goblet cells, which forms a positive feedback loop with tuft cells and promotes the mucus secretion to expel parasites from the intestine[18,19] (Fig. 1a).

Here, we found that p53 is essential for the intestinal type 2 immunity in response to parasitic infections; p53 deficiency results in an impaired intestinal type 2 innate immunity towards the infection of protozoa and helminth parasites. Mechanistically, p53 transcriptionally upregulates Lrmp, a previously unrecognized p53 target gene, which plays an important role in ensuring that intestinal tuft cells trigger the type 2 immunity in response to parasitic infections. This study unveils an important role of p53 in innate immunity in the context of parasitic infections.

## Results

### p53 deficiency impairs parasite-induced type 2 immune response

To investigate whether p53 regulates the type 2 immune response towards parasitic infections, p53+/+ and p53−/− mice were infected with a non-pathogenic protozoa Tritrichomonas muris (Tm) (Fig. S1a) by oral gavage. The hyperplasia of tuft and goblet cells in the intestine, the features of type 2 immune response, were analyzed in the intestine from the infected mice at 21 days post-infection (d.p.i.). Tuft cells were detected by immunohistochemistry (IHC) staining of Dclk1, a tuft cell marker[20], and goblet cells were detected by Alcian blue staining[21], respectively. Tm infection clearly induced hyperplasia of both tuft and goblet cells in the small intestine in p53+/+ mice, which was significantly blunted in p53−/− mice (Fig. 1b, c). Tm infection also induced the activation and expansion of ILC2s, another feature of type 2 immune response, in the LP of p53+/+ mice as determined by flow cytometric analysis (Fig. 1d). Further, eosinophilia was observed in the LP of Tm-infected p53+/+ mice as determined by flow cytometric analysis and H&E staining, respectively (Fig. 1d, Fig. S1b). Compared with p53+/+ mice, the increase of ILC2 numbers was much less pronounced and eosinophilia was very limited in the LP of Tm-infected p53−/− mice (Fig. 1d, Fig. S1b). ILC2s are a robust intestinal source of IL-13[22,23]. Compared with p53+/+ naïve mice, the levels of IL-13 mRNA in the intestinal tissues were significantly increased in Tm-infected p53+/+ mice as determined by quantitative real-time PCR assays (Fig. 1e). Notably, compared with p53+/+ mice, the increase in IL-13 mRNA levels in the intestinal tissues was much less pronounced in Tm-infected p53−/− mice (Fig. 1e). Furthermore, Tm infection significantly increased the levels of serum IL-4, another type 2 cytokine, in Tm-infected p53+/+ mice as determined by ELISA assays (Fig. S1c). Compared with p53+/+ mice, the increase in serum IL-4 levels was much less pronounced in Tm-infected p53−/− mice. Notably, p53−/− mice had a significantly higher Tm burden in the cecum than p53+/+ mice at 21 d.p.i. (Fig. 1f). Further, the uninfected p53+/+ and p53−/− mice had comparable levels of tuft and goblet cells, eosinophils, ILC2s, and IL-13 in the intestine, as well as the serum IL-4 levels (Fig. 1b–e, Fig. S1b, c).

Tm is a non-pathogenic protozoa and a stable component of the microbiota. Here, we further investigated whether p53 regulates the type 2 immune response towards the infection of Nippostrongylus brasiliensis (Nb), a pathogenic helminth (Fig. S1d). Nb induces a strong type 2 immune response that clears intestinal worms 7–10 d.p.i.[17]. A dramatic hyperplasia of tuft and goblet cells in the small intestine was observed in Nb-infected p53+/+ mice at 7 d.p.i., which was significantly blunted in Nb-infected p53−/− mice (Fig. 1g, h). The significantly higher number of eosinophils and ILC2s in the LP and mesenteric lymph nodes (MLNs) (Fig. 1i, Fig. S1e, f), and significantly higher levels of IL-13 and IL-4 (Fig. 1j, Fig. S1g) were observed in Nb-infected p53+/+ mice than p53−/− mice. Notably, the Nb-infected p53−/− mice exhibited an impaired ability to expel Nb; p53−/− mice had a significantly higher number of eggs in the feces and a significantly higher worm burden in the intestine than p53+/+ mice at 7 d.p.i. (Fig. 1k). Collectively, these results demonstrate that p53 is required for the type 2 innate immune response to protect mice from the infection of protozoa and helminth parasites.

### p53 deficiency impairs the type 2 immune response induced by succinate

Recent studies have shown that the microbial metabolite succinate secreted by Tm and Nb is an activating ligand for intestinal tuft cells[24,25]. The succinate receptor (SUCNR1), which is specifically expressed in intestinal tuft cells, detects succinate and leads to the activation of the tuft-ILC2 circuit[24]. It has been reported that dietary succinate in drinking water can trigger the tuft-ILC2 circuit, leading to the hyperplasia of intestinal tuft and goblet cells, eosinophilia, and the increase of ILC2 numbers in the

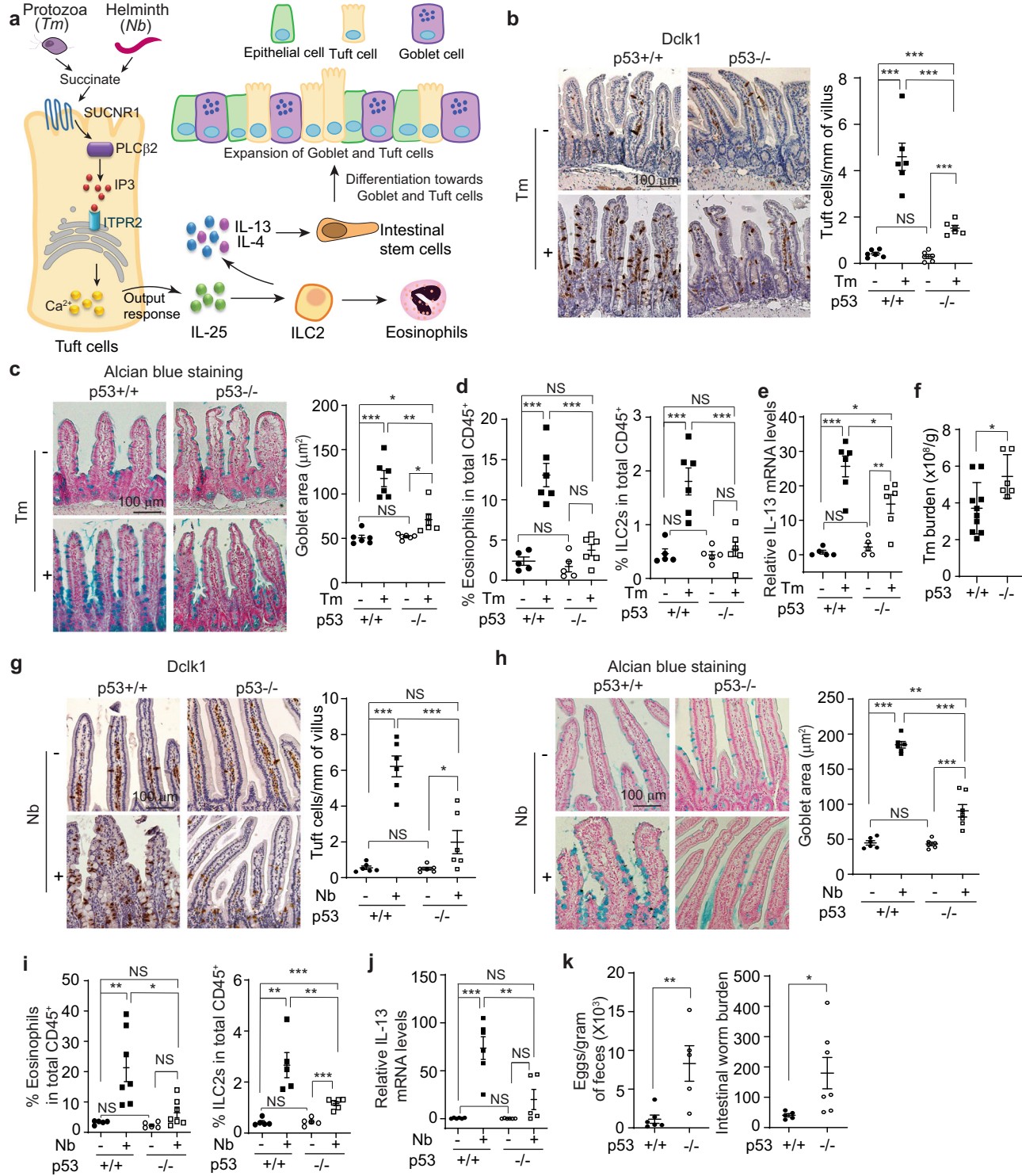

small intestine[24,25]. Indeed, providing succinate in drinking water induced the intestinal type 2 immune response in *p53+/+* mice; succinate induced hyperplasia of tuft and goblet cells, induced eosinophilia, increased ILC2 numbers, and increased the levels of IL-13 and IL-4 in *p53+/+* mice (Fig. 2a–c, Fig. S2). Notably, loss of p53 greatly impaired succinate-induced intestinal type 2 immune response (Fig. 2a–c, Fig. S2). Compared with succinate-treated *p53+/+* mice, the hyperplasia of tuft and goblet cells and the increase of ILC2 numbers and eosinophilia were much less pronounced in succinate-treated *p53−/−* mice (Fig. 2a, b, Fig. S2a, b). Further, succinate treatment did not have a significant

effect on the levels of IL-13 and IL-4 in *p53−/−* mice (Fig. 2c, Fig. S2c). These results further support that p53 is required for intestinal type 2 innate immunity.

**p53 deficiency impairs the function of tuft cells to release IL-25.** In response to parasitic infections, tuft cells release IL-25 to trigger type 2 immunity[15–17]. Here, we investigated whether the impaired type 2 immune response towards parasitic infections in *p53−/−* mice is due to the impaired function of tuft cells. Infections of *Tm* and *Nb* or succinate treatment increased IL-25 to a much higher level in the intestinal epithelium of *p53+/+*

**Fig. 1 The type 2 immune response induced by parasitic infections is impaired in _p53−/−_ mice. a** Schematic illustration of the signaling pathway utilized by tuft cells to initiate the type 2 immunity towards parasitic infections. **b–k** _p53+/+_ and _p53−/−_ mice were infected with _Tritrichomonas muris_ (_Tm_) by oral gavage (_P.O._) and examined at 21 d.p.i. **b–f** or with _Nippostrongylus brasiliensis_ (_Nb_) by subcutaneous (_s.c._) injection and examined at 5 d.p.i. for flow cytometric analysis and 7 d.p.i. for other assays **g–k**. **b** and **g** Tuft cell lineage expansion was assessed by Dclk1 IHC staining in the small intestine of naïve and infected mice. Left: representative images. Right: quantifications of tuft cell number in the villi. **c** and **h** Goblet cell hyperplasia was assessed by Alcian blue staining in the small intestine of naïve and infected mice. Left: representative images. Right: quantifications of the goblet area. **d** and **i** Flow cytometric analysis of the population of eosinophils and ILC2s in the lamina propria (LP). Flow cytometry gating strategy and representative images are shown in Fig S11. **e** and **j** Relative mRNA levels of IL-13 in the small intestine of naïve and infected mice as determined by quantitative real-time PCR and normalized with β-actin. **f** _Tm_ numbers in cecum at 21 d.p.i. in _p53+/+_ and _p53−/−_ mice. **k** Egg counts in feces and worm burden in the intestine at 7 d.p.i. in _p53+/+_ and _p53−/−_ mice. For **b–e**, **g–j**, the ileum, and duodenum tissues were collected from _Tm_-infected and _Nb_-infected mice, respectively, for analysis. **b–k** Data are presented as mean ± SEM. Each dot represents an individual mouse. $n = 5–7$/group. For **b** and **g** ≥30 villus/mouse were counted. For **c** and **h**, ≥50 goblet cells/mouse were quantified. *$p < 0.05$; **$p < 0.01$; ***$p < 0.001$; NS: non-significant, two-tailed Student's _t_-test.

mice than that of _p53−/−_ mice (Fig. 2d, Fig. S3a). Further, recombinant IL-25 (rIL-25) was administrated to mice to mimic the release of IL-25 by tuft cells, and the type 2 immune response towards rIL-25 treatment was examined in _p53+/+_ and _p53−/−_ mice. Administering rIL-25 (_i.p._, 0.5 μg/day for 7 days) to mice induced hyperplasia of tuft and goblet cells, increased the population of eosinophils and ILC2s, and increased the levels of IL-13 and IL-4 to a similar extent in _p53+/+_ and _p53−/−_ mice (Fig. 2e–g, Fig. S3b–d), suggesting that p53 loss impairs the function of tuft cells in response to parasitic infections, which can be restored by administration of rIL-25. IL-25 promotes the expansion of ILC2s, which are a critical source of IL-13, a major type 2 cytokine playing a key role in tuft cell hyperplasia[18,19]. Here, we compared IL-13-driven tuft cell hyperplasia between _p53+/+_ and _p53−/−_ intestinal organoids, which reconstitute all intestinal epithelial cell types, by treating intestinal organoids with the recombinant IL-13 (rIL-13). The rIL-13 induced tuft cell expansion in _p53+/+_ and _p53−/−_ intestinal organoids to a similar extent (Fig. 2h). Collectively, these data indicate that p53 is required for tuft cells to trigger the type 2 immunity in response to parasitic infections; loss of p53 impairs the type 2 immunity in response to parasitic infections and succinate treatment, which can be restored by administering rIL-25 in mice.

**Lrmp is a p53 target gene highly expressed in tuft cells**. As a transcription factor, p53 mainly exerts its function through transcriptional regulation of its target genes by binding to the p53-binding elements in the regulatory region of its target genes[3,4]. Here, we investigated whether p53 regulates tuft cell function in the tuft cell-ILC2 axis through transcriptional regulation of its target gene(s) in tuft cells. The p53MH algorithm is a computational program developed for genome-wide scanning for potential p53 targets by identifying putative p53-binding elements in genes[26], which has been used to successfully identify many p53 target genes[27,28]. Here, the p53MH was employed to search for potential p53 target genes expressed in the intestinal tuft cells by analyzing over 100 tuft cell signature genes defined by the single-cell RNA (scRNA) expression dataset of mouse small intestinal epithelium (GSE92332)[29]. The _Lrmp_ gene was identified as a potential p53 target gene with two putative p53-binding elements in the intron 1 of the mouse _Lrmp_ gene (Fig. 3a). Interestingly, a previous report using the hidden variable dynamic modeling (HVDM) approach also predicted Lrmp as a putative p53 target gene[30]. Chromatin immunoprecipitation (ChIP) assays were performed in mouse Val5 fibroblasts containing a temperature-sensitive mutant p53 vector, which expresses wild-type (WT) p53 protein at 32 °C and loss-of-function mutant p53 protein at 37 °C. Immunoprecipitation of the chromatin fragments corresponding to both potential p53-binding elements in the mouse _Lrmp_ gene with an anti-p53 antibody was observed only when WT p53 protein was expressed at 32 °C, indicating

that WT p53 binds to these two potential p53-binding elements in vivo (Fig. 3b). Further, pGL2 luciferase reporter vectors containing these two potential p53-binding elements exhibited a p53-dependent transcriptional activation in _p53_-null MEFs cotransfected with WT p53 or control vectors (Fig. 3c). Notably, culturing cells at 32 °C, which induced WT p53 expression in Val5 cells, significantly induced _Lrmp_ expression in Val5 cells but not in 10(1) cells, the _p53_-null parental cells of Val5 (Fig. 3d). Furthermore, the basal _Lrmp_ mRNA levels were significantly higher in _p53+/+_ MEFs than _p53−/−_ MEFs, suggesting that p53 has a significant impact on the basal _Lrmp_ transcription levels (Fig. 3d). These data reveal that _Lrmp_ is a previously unidentified p53 target gene, and p53 regulates both basal and inducible transcription of _Lrmp_.

_Lrmp_ is highly expressed in intestinal tuft cells. Mining scRNA expression dataset of mouse small intestinal epithelium (GSE92332)[29] showed that _Lrmp_ expression was detected in 116 cells out of 1522 epithelial cells with full-length scRNA-seq (Fig. 3e). Notably, 87% of _Lrmp_-expressing cells were tuft cells ($n = 101$), which accounts for 99% (101 out of 102 cells) of the tuft cells sequenced in this dataset. The average _Lrmp_ expression levels in these tuft cells (2100 ± 1188) were ~5–100-fold higher than the remaining 15 _Lrmp_-expressing non-tuft cells (14 ± 19 in enterocytes, $n = 3$; 20 ± 23 in paneth cells, $n = 8$; 22 ± 29 in stem cells, $n = 2$; and 411 ± 555 in goblet cells, $n = 2$) (Fig. 3e). Dclk1 is a well-established tuft cell marker[20]. Immunofluorescence (IF) staining of mouse intestinal organoids showed that Lrmp was specifically expressed in Dclk1-positive epithelial cells (Fig. 3f), confirming that Lrmp is specifically expressed at high levels in tuft cells. Similar results were obtained with IF staining of mouse intestinal tissues; positive staining of Lrmp was observed in 98.3% of tuft cells with Dclk1-positive staining (294 out of 299 cells) whereas no obvious Lrmp staining was observed in paneth cells with positive staining of lysozyme, a marker for paneth cells (0 out of 400 cells) (Fig. S4a). It has been well-established that tuft cells can be enriched by sorting EpCAM$^+$SiglecF$^+$ population from intestinal single-cell suspensions[16]. Analysis of Lrmp expression in the enriched tuft cells by sorting EpCAM$^+$SiglecF$^+$ population from intestinal single-cell suspensions showed that _p53+/+_ tuft cells displayed significantly higher levels of Lrmp (Fig. 3g) but similar levels of Dclk1 compared with _p53−/−_ tuft cells (Fig. S4b). IF staining of intestinal organoids showed that the mean fluorescence intensity (MFI) of Lrmp was significantly higher in _p53+/+_ tuft cells than _p53−/−_ tuft cells (Fig. S4c). _Tm_ and _Nb_ infections, as well as succinate treatment, increased p53 protein levels in tuft cells as determined by IF co-staining of Dclk1 and p53 in intestinal tissues (Fig. S4d). Furthermore, while p53 activated by γ-irradiation induces the levels of p21 and MDM2, two classical p53 targets, in the intestinal epithelium, no obvious increase of the levels of p21 and MDM2 was observed in the tuft cells from mice with _Tm_ and _Nb_ infections or succinate treatment

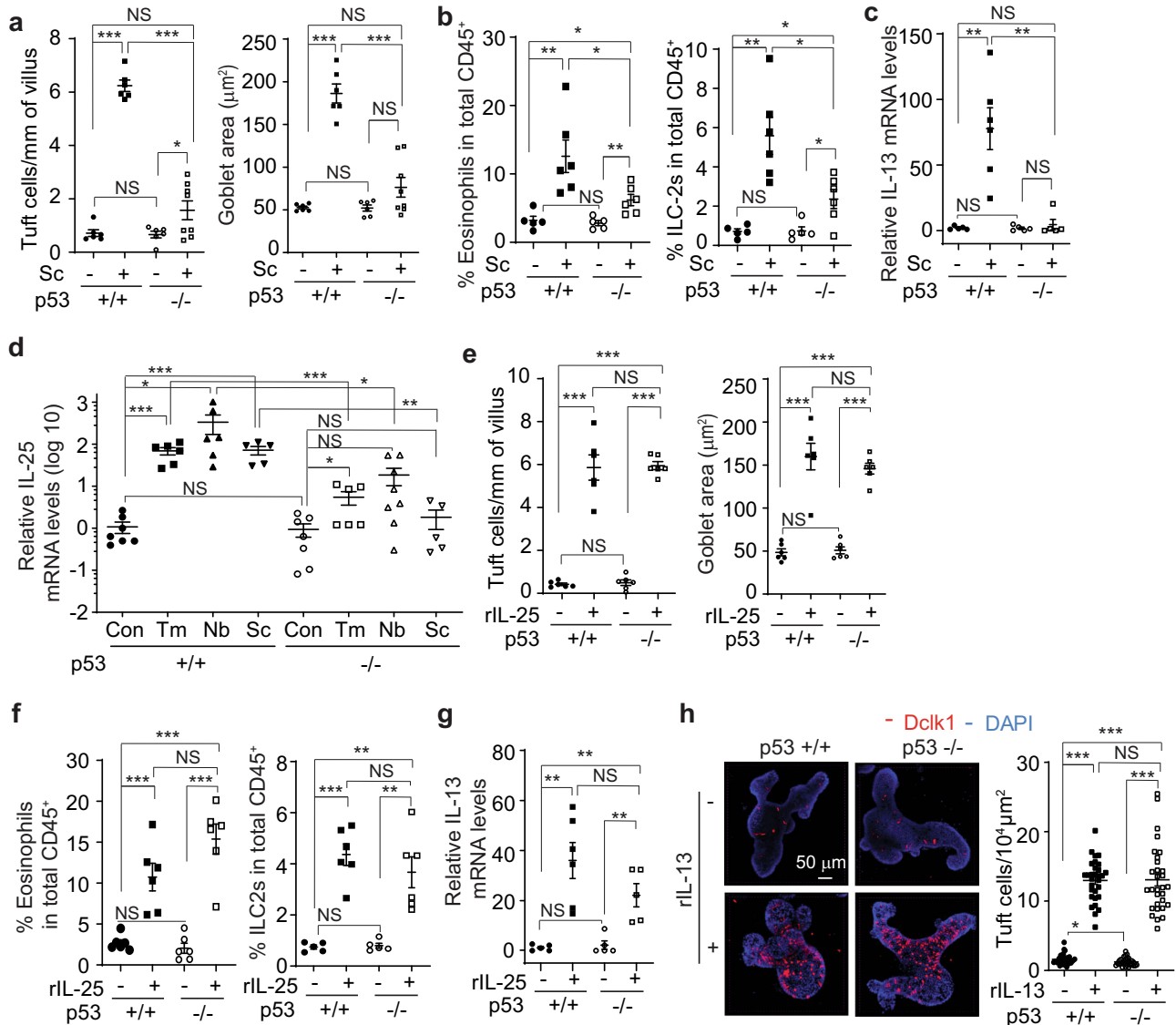

**Fig. 2 p53 deficiency impairs the function of tuft cells to trigger type 2 immunity. a–c** *p53+/+* and *p53−/−* mice were given drinking water with or without 150 mM succinate (Sc) for 7 days before analysis. **a**. Quantifications of tuft (left) and goblet cells (right) in the small intestine of mice treated with or without succinate by IHC staining of Dclk1 and Alcian blue staining, respectively. **b** Quantifications of eosinophils and ILC2s in the LP by flow cytometric analysis. **c** Relative mRNA levels of IL-13 in the small intestine. In **c**, **d**, and **g**, mRNA levels were determined by quantitative real-time PCR and normalized with β-actin. **d** Relative mRNA levels of IL-25 in the intestinal epithelium of naïve mice, mice infected with *Tm* or *Nb*, and mice treated with succinate. **e–g** *p53+/+* and *p53−/−* mice were treated with rIL-25 (0.5 μg/day; *i.p.*) or PBS for 7 days before analysis. **e** Quantifications of tuft (left) and goblet cells (right) in the small intestine. **f** Quantifications of eosinophils and ILC2s in the LP. **g** Relative mRNA levels of IL-13 in the small intestine. **h** rIL-13 induced tuft cell expansion in *p53+/+* and *p53−/−* intestinal organoids to a similar extent. The intestinal organoids from *p53+/+* and *p53−/−* mice were treated with rIL-13 (10 ng/ml for 48 h) or PBS, and then tuft cells were detected by IF staining of Dclk1. Left: representative images. Right: quantifications of tuft cells. For **a–h**, data are presented as mean ± SEM. For **a–g**, each dot represents an individual mouse. *n* = 5–8/group. For **h**, each dot represents an image field. *p < 0.05; **p < 0.01; ***p < 0.001; NS: non-significant, two-tailed Student's *t*-test.

as determined by the IF staining (Fig. S4e). Together, these results demonstrate that p53 selectively upregulates *Lrmp* transcription to maintain its high expression in tuft cells, and loss of p53 reduces Lrmp levels in tuft cells.

**Lrmp interacts with ITPR2 and regulates Ca$^{2+}$ flux in cells.** Intestinal tuft cells encode a chemo-sensing pathway that was previously characterized in taste transduction[31]. Chemo-sensing by intestinal tuft cells regulates the tuft-ILC2 circuit[24,25,32]. The activation of surface-expressed G protein-coupled taste receptors (GPCRs) by their ligands activates Plcβ2 to generate the second messenger inositol 1,4,5-triphosphate (IP3), which binds to its

receptor ITPR2 in the endoplasmic reticulum (ER) to trigger Ca$^{2+}$ release from the ER (Fig. 1a)[32]. ITPR2 is the major receptor of the IP3 receptor (IP3R) family expressed in the intestinal tuft cells[32]. The elevation of intracellular Ca$^{2+}$ levels opens the transient receptor potential cation channel subfamily M member 5 (Trpm5), which results in IL-25 release from tuft cells to modulate the intestinal type 2 immune response (Fig. 1a)[32]. Lrmp was identified as a transmembrane protein localized on the cytoplasmic face of the ER in cells[33]. To elucidate the role of Lrmp in the tuft-ILC2 circuit, we searched for potential Lrmp-interacting proteins in MEFs with ectopic Lrmp-Flag expression using co-immunoprecipitation (co-IP) assays with an anti-Flag

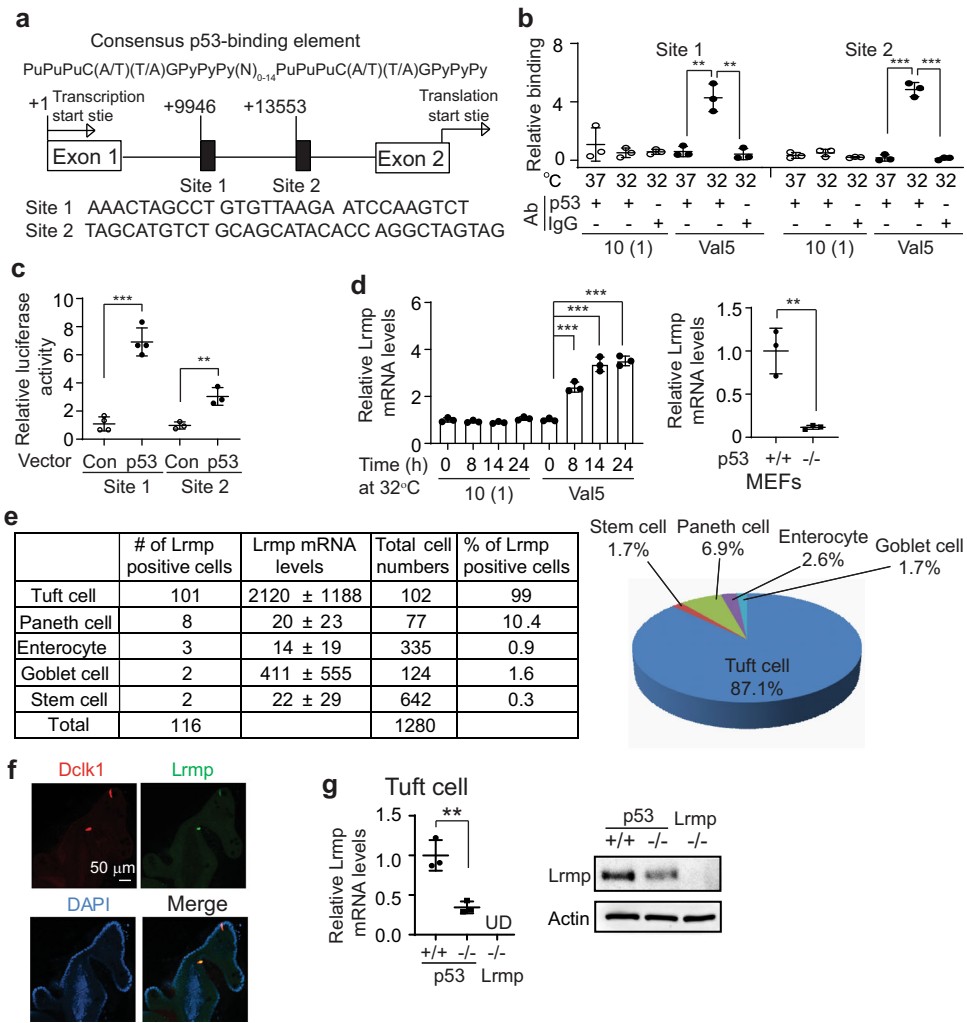

**Fig. 3 Lrmp is a p53 target gene highly expressed in tuft cells. a** The putative p53-binding elements in the *Lrmp* gene predicted by the p53MH program. A consensus p53-binding element is also presented. Pu, purine; Py, pyramidine; N, any nucleotide. **b** p53 bound to two p53-binding elements in *Lrmp* determined by ChIP assays. **c** Ectopic expression of p53 transactivated the p53-binding elements in *Lrmp* determined by luciferase reporter assays in *p53−/−* MEFs. **d** p53 transcriptionally upregulated *Lrmp* expression. mRNA levels were determined by quantitative real-time PCR and normalized with β-actin. **e** *Lrmp* expression in mouse intestinal epithelial cells by analyzing a scRNA-seq dataset (GSE92332). **f** Co-localization of Lrmp and Dclk1 in tuft cells determined by IF staining of intestinal organoids. **g** *Lrmp* mRNA and protein levels in *p53+/+*, *p53−/−* and *Lrmp−/−* tuft cells determined by quantitative real-time PCR and Western-blot assays, respectively. For **b**, **c**, **d**, and **g**, data are presented as mean ± SD. **$p < 0.01$; ***$p < 0.001$, two-tailed Student's *t*-test. UD: undetectable.

antibody followed by chromatography-tandem mass spectrometry (LC-MS/MS) assays. Three IP3Rs, including ITPR1, ITPR2 and ITPR3, were identified as potential Lrmp-binding proteins in the chemo-sensing pathway as analyzed by the KEGG mapper (Fig. 4a, b). Analysis of the expression levels of IP3Rs in intestinal tuft cells using the scRNA-seq dataset (GSE92332)[29] confirmed that ITPR2 is the major IP3R expressed in intestinal tuft cells (Fig. S5). Therefore, MEFs co-transfected with vectors expressing Lrmp-Flag and ITPR2-HA, respectively, were subjected to co-IP using an anti-Flag antibody followed by LC-MS/MS and Western-blot assays, respectively. The Lrmp-ITPR2 interaction was observed in both assays (Fig. 4b, c). To determine the region in Lrmp required for its interaction with ITPR2, vectors expressing a series of deletion mutants of Lrmp-Flag and WT ITPR2-HA, respectively, were co-transfected into human H1299 cells for co-IP assays (Fig. 4d). H1299 cells were used for their high efficiency for vector transfections. We found that the middle region (aa 200–340) which contains the coiled-coil domain (aa 264–291) was required for Lrmp to bind to ITPR2; ITPR2 interacted with

the full-length Lrmp (FL-Lrmp) and Lrmp fragments that contain middle region (aa 200–340), but not the fragment without the middle region (ΔM) (Fig. 4d). The direct interaction between Lrmp and ITPR2 was further confirmed in WT small intestinal tissues by employing the Proximity Ligation Assays (PLAs), an assay widely used for detecting direct protein-protein interactions in vivo[34]; the Lrmp-ITPR2 interaction was observed in the tuft cells (identified by the IF staining of Dclk1) in WT but not *Lrmp−/−* small intestinal tissues (Fig. 4e). Given the critical role of ITPR2 in regulating the Ca²⁺ flux, the observation of the Lrmp-ITPR2 interaction raises the possibility that Lrmp modulates Ca²⁺ flux in cells. Lrmp knockout (*Lrmp−/−*) mice were generated by crossing Lrmp^tm1a(EUCOMM)Wtsi/WtsiOulu mice (EUCOMM) with E2a-Cre mice, in which Cre is expressed in the germline[35], and then backcrossed with C57BL6/J for at least 5 generations (Fig. S6a–c). *Lrmp−/−* mice were viable and showed no obvious abnormality. The deficiency of Lrmp expression was confirmed at both mRNA and protein levels in intestinal tuft cells (Fig. 3g, Fig. S4f). *Lrmp−/−* and WT MEFs were employed to

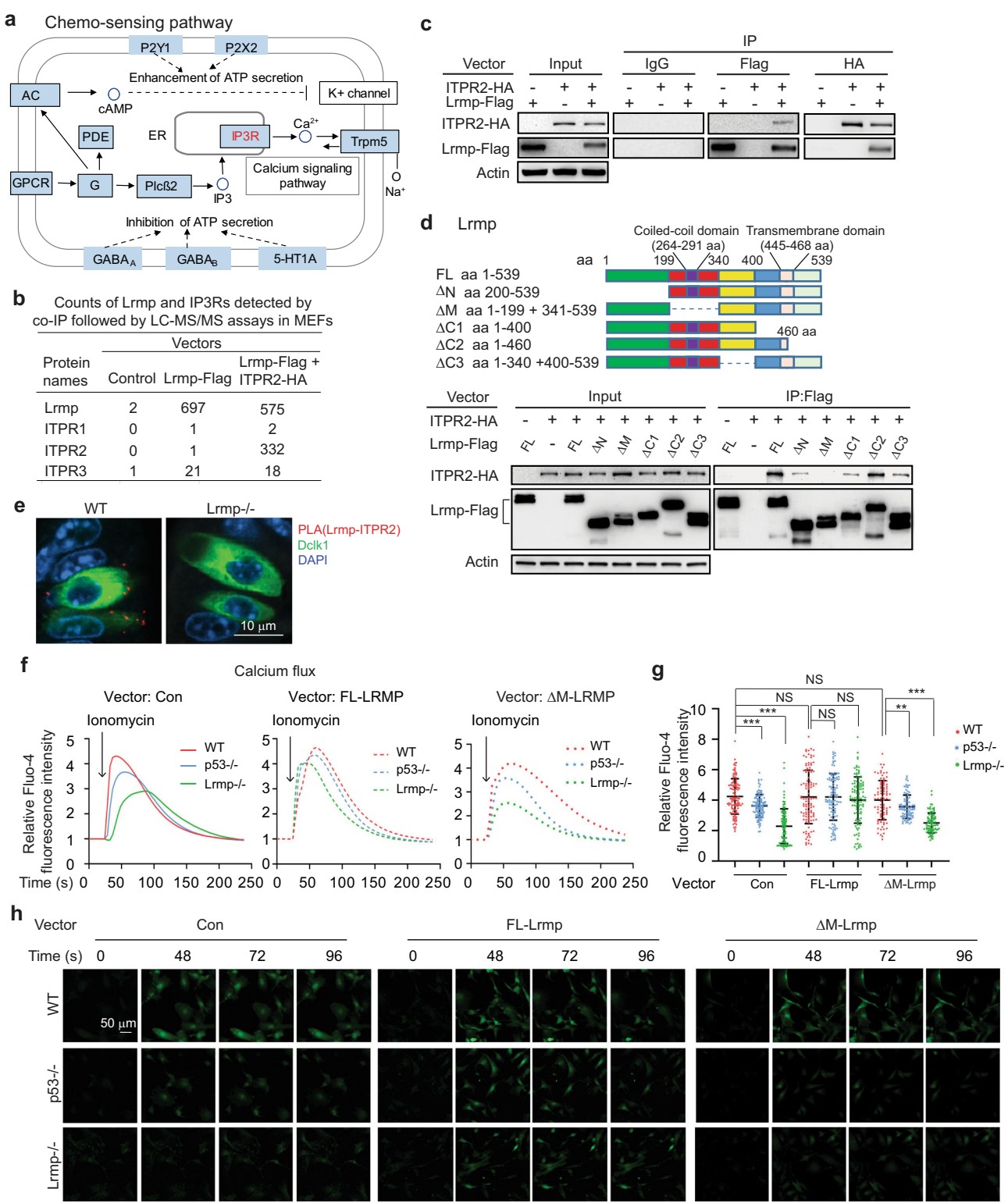

determine Ca²⁺ release from the ER induced by Ca²⁺ ionophore ionomycin, which requires the normal function of IP3Rs[36]. Intracellular Ca²⁺ influx was measured by using Fluo-4 AM with a time-lapse confocal microscope. Ionomycin treatment significantly increased the relative intensity of Fluo-4 AM in WT MEFs, and this effect was greatly reduced in *Lrmp*−/− MEFs (Fig. 4f–h, Fig. S7). Notably, the impaired ionomycin-induced Ca²⁺ influx in *Lrmp*−/− MEFs was largely rescued by ectopic expression of FL-Lrmp in the MEFs (Fig. 4f–h, Fig. S7).

Furthermore, compared with *p53+/+* MEFs, ionomycin-induced Ca²⁺ influx was greatly reduced in *p53*−/− MEFs, which expressed much lower levels of Lrmp (Fig. 4f–h, Fig. S7). Ectopic expression of FL-Lrmp in *p53*−/− MEFs clearly enhanced ionomycin-induced Ca²⁺ influx, but showed a much less pronounced effect on ionomycin-induced Ca²⁺ influx in *p53+/+* MEFs (Fig. 4f–h, Fig. S7). In contrast, ectopic expression of ΔM-Lrmp, which does not interact with ITPR2, had no obvious effect on ionomycin-induced Ca²⁺ influx in *p53+/+*, *p53*−/−, or

**Fig. 4 Lrmp interacts with ITPR2 and regulates Ca$^{2+}$ flux in cells. a** and **b** Lrmp interacted with IP3R proteins, including ITPR2, as determined by co-IP followed by LC-MS/MS assays in MEFs. **a** Lrmp-interacting proteins (label with red) mapped in "Chemo-sensing pathway" in KEGG. **b** Counts of Lrmp and IP3Rs detected by co-IP followed by LC-MS/MS assays in MEFs. **c** Lrmp-ITPR2 interaction was confirmed by co-IP assays followed by Western-blot assays in MEFs transfected with vectors expressing Lrmp-Flag and ITPR2-HA, respectively. **d** The binding region of Lrmp for ITPR2 was determined by co-IP followed by Western-blot assays in H1299 cells transfected with vectors expressing different Lrmp-Flag fragments and ITPR2-HA, respectively. Upper panel: schematic representation of vectors expressing full-length (FL) or serial deletion mutants of Lrmp-Flag. **e** The Lrmp-IPTR2 interaction was examined by PLA assays in WT and *Lrmp−/−* small intestinal tissues. Red: PLA signals of the Lrmp-ITPR2 interaction; green: Dclk1; DAPI: nucleus. **f–h** Ca$^{2+}$ flux in response to Ionomycin (2 μM) treatment in WT, *p53−/−* and *Lrmp−/−* MEFs with or without transfection of either FL- or ΔM-Lrmp-Flag vectors. **f** Average traces of Ca$^{2+}$ responses. **g** Relative Fluo-4 fluorescence obtained from Ca$^{2+}$ transients at peak (48 s). **h** Representative fluorescence images. The Fluo-4 fluorescence intensity before Ionomycin treatment was calculated as 1. In **f**, each curve represents an average of at least 90 cells. Curves of each cell are shown in Fig S7. ***$p < 0.001$; **$p < 0.01$; NS: non-significant, two-tailed Student's *t*-test.

*Lrmp−/−* MEFs (Fig. 4f–h, Fig. S7). Taken together, these results suggest that p53 ensures Ca$^{2+}$ flux in tuft cells through upregulation of Lrmp levels.

**The impaired intestinal type 2 immune response in *Lrmp−/−* mice.** We further investigated whether Lrmp mediates p53's function in intestinal tuft cell-IL-25-ILC2 circuit activation in response to parasitic infections. Compared with their WT littermates, intestinal tuft cell-IL-25-ILC2 circuit activation was greatly impaired in *Lrmp−/−* mice infected with *Tm* or *Nb*. The hyperplasia of tuft and goblet cells in the small intestine was observed in WT mice but not observed or much less pronounced in the infected *Lrmp−/−* mice (Fig. 5a, b, Fig. S8a, b). The clear increase in the numbers of eosinophils and ILC2s and the induction of the levels of IL-13 and IL-4 were observed in the infected WT mice but not observed or much less pronounced in the infected *Lrmp−/−* mice (Fig. 5c–e, Fig. S8c–e). Furthermore, Lrmp deficiency in mice resulted in a significantly impaired ability to expel *Nb* from the gut; *Lrmp−/−* mice had a significantly higher number of eggs in the feces and a high worm burden in the intestine than WT mice at 7 d.p.i. (Fig. 5f).

We further examined the type 2 immune response towards succinate treatment in *Lrmp−/−* mice. *Lrmp−/−* mice displayed an impaired type 2 immune response towards succinate treatment; compared with WT mice, succinate treatment resulted in a much weaker promoting effect on the hyperplasia of tuft and goblet cells, the increase in the numbers of eosinophils and ILC2s, and the induction of the levels of IL-13 and IL-4 in *Lrmp−/−* mice (Fig. 5g–i, Fig. S9a–c). These results demonstrate that Lrmp deficiency in mice significantly impairs type 2 immune response towards the infections of protozoa and helminth parasites and succinate treatment.

**Recombinant IL-25 rescues the type 2 immune response in *Lrmp−/−* and *p53−/−* mice.** Ca$^{2+}$ flux is critical for the release of the cytokine IL-25 from tuft cells to trigger the type 2 immune response[32]. Given the important role of Lrmp in modulating Ca$^{2+}$ flux in cells, we examined the levels of IL-25 in *Lrmp−/−* mice in response to parasitic infections and the succinate treatment. Compared with WT mice, the infection of *Tm* and *Nb* or succinate treatment led to a much less pronounced increase in IL-25 levels in the intestinal epithelium of *Lrmp−/−* mice, which was similar to the observation made in *p53−/−* mice (Fig. 6a, Fig. S10a). To investigate whether the impaired release of IL-25 from tuft cells in response to parasitic infections leads to the impaired type 2 immune response in *Lrmp−/−* mice, the type 2 immune response was examined in *Lrmp−/−* mice administered with rIL-25. Intriguingly, rIL-25 treatment (*i.p.*, 0.5 μg/day for 7 days) effectively induced type 2 immune response in *Lrmp−/−* mice to a similar extent as that in WT mice, including hyperplasia of tuft and goblet cells, the increases in the numbers of eosinophils,

ILC2s and the levels of IL-13 and IL-4 (Fig. 6b–d, Fig. S10b-d). Consistent with these results in mice, rIL-13 treatment induced tuft cell expansion to a similar extent in WT and *Lrmp−/−* organoids (Fig. S10e).

We further investigated whether rIL-25 treatment rescues the impaired ability to expel *Nb* in *p53−/−* and *Lrmp−/−* mice. Mice infected with *Nb* were injected with rIL-25 (*i.p.*, 0.5 μg once every 2 days for 7 days), and *Nb* egg counts in the feces and worm burden in the intestine were determined at 7 d.p.i. The rIL-25 treatment greatly increased the resistance of mice towards *Nb* infection, and largely rescued the impaired ability to expel *Nb* in *p53−/−* and *Lrmp−/−* mice; *p53+/+*, *p53−/−*, and *Lrmp−/−* mice with rIL-25 treatment showed similar levels of egg counts in the feces and worm burden in the intestine at 7 d.p.i. (Fig. 6e). Collectively, these data demonstrate that Lrmp, a p53 target gene, is important for tuft cells to respond to the infections of protozoa and helminth parasites and trigger intestinal type 2 immunity, and suggest that Lrmp is an important mediator for the role of p53 in regulating type 2 immunity in response to parasitic infections (Fig. 6f).

## Discussion

In 1979, p53 was discovered as a cellular protein binding to SV40 large T antigen, the oncoprotein of SV40 virus[5,6]. Since then, extensive studies on p53 have unequivocally established its central role in tumor suppression. The loss of p53 is often a prerequisite for the development and progression of many human tumors[2,3,37,38]. The p53 gene is the most frequently mutated gene in human cancers; approximately 50% of all human cancers contain a mutation in the *p53* gene. p53 senses diverse stress-related stimuli and selectively regulates transcription of its target genes to induce different biological outcomes in a cell, tissue, and stress type-specific manner, including apoptosis, cell cycle arrest, senescence, DNA repair, cell metabolism, and antioxidant defense, to safeguard the integrity of the genome[2,3,39].

While the majority of research on p53 has been focused on cancer biology, accumulating evidence has suggested the involvement of p53 in immune function. In response to the infections of many different viruses, p53 is activated and in turn kills the virus-infected cells to limit viral replication, which demonstrates the role of p53 in the innate immune response towards viral infections[7]. It is therefore not surprising that p53 can be inactivated by many viruses to prevent host cell-mediated apoptosis and allow the replication of viruses[40,41]. p53 can induce the transcription of some members of Toll-like receptors (TLRs), including TLR3 and TLR9[42,43]. TLRs recognize pathogen-associated and damage-associated molecular patterns to promote a protein kinase cascade that induces the expression of inflammatory cytokines and interferons. Increased expression of these TLRs induced by p53 can trigger agonist-induced apoptotic cell death[42,43]. p53 also connects to innate immunity through its regulation of cellular senescence. Cellular senescence is induced

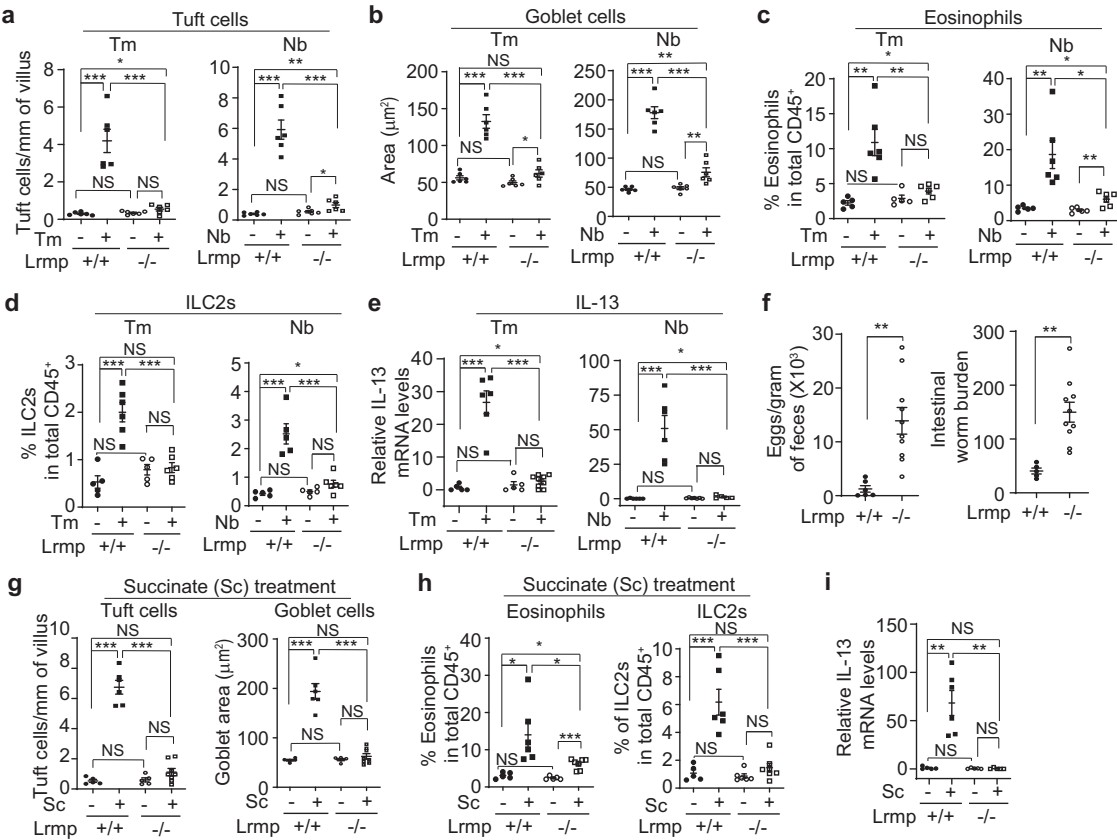

**Fig. 5 The intestinal type 2 immune response induced by parasitic infections and succinate is impaired in *Lrmp−/−* mice. a–f** WT and *Lrmp−/−* mice were infected with *Tm* or *Nb* and examined at 21 d.p.i. (for *Tm*) and 5 or 7 d.p.i. (for *Nb*), respectively. **a** and **b** Quantifications of tuft (**a**) and goblet (**b**) cells in the small intestine of mice by IHC staining of Dclk1 and Alcian blue staining, respectively. **c** and **d** Quantifications of eosinophils (**c**) and ILC2s (**d**) in the LP by flow cytometric analysis. **e** Relative IL-13 mRNA levels in the small intestine determined by quantitative real-time PCR assays. **f** Egg counts in feces (left) and worm burden (right) in the intestine in *Nb*-infected mice. **g–i** The impaired type 2 immune response induced by succinate (150 mM for 7 days) in *Lmrp−/−* mice. **g** Quantifications of tuft and goblet cells in the small intestine. **h** Quantifications of eosinophils and ILC-2s in the LP. **i** Relative IL-13 mRNA levels in the small intestine. Data are presented as mean ± SEM. Each dot represents an individual mouse. $n = 5$–8/group. $*p < 0.05$; $**p < 0.01$; $***p < 0.001$; NS: non-significant, two-tailed Student's *t*-test.

in stressed and damaged cells as a strategy to prevent the proliferation of damaged cells and to maintain tissue integrity[44]. p53 is an important inducer of senescence[45]. In addition to stopping cellular growth, senescence triggers the production and secretion of an array of chemokines and cytokines, which attract natural killer (NK) cells, CD-8 killer T cells, and monocytes/macrophages to kill senescent cells and clean up the cellular debris, known as senescence-associated secretory phenotype (SASP)[44]. While we began to appreciate the connection between p53 and the immune system, the precise role of p53 in the immune response under different conditions is far from clear.

This study reveals a previously unrecognized function of p53 in the regulation of the intestinal type 2 innate immune response to protect against the infection of protozoa and helminth parasites. We found that p53 is crucial for the function of intestinal tuft cells to trigger type 2 immunity in response to parasitic infections. Tuft cells are an epithelial lineage that has typical bottle-shaped and apical microvilli morphology and uses key constituents of the taste receptor signaling cascade[46]. The functions of intestinal tuft cells were not clear for over 50 years since their discovery until recent reports revealed their key role in sensing parasites and inducing the type 2 immune response to protect hosts from parasitic infections[15–17]. Intestinal tuft cells can sense parasitic infections using metabolic and taste-activated GPCRs. For example, SUCNR1 can sense succinate produced by *Tm* and some helminth parasites, and taste-activated GPCRs can sense

*Trichinella spiralis*, a helminth parasite[24,32]. Ligand binding to GPCRs induces the release of the Gα protein from the trimeric G protein complex, which in turn activates PLCβ2-mediated cleavage of phosphatidylinositol-4,5-bisphosphate (PIP2) into IP3. IP3 binds to IP3Rs on the ER to release intracellular $Ca^{2+}$ stores into the cytosol, which in turn triggers TRPM5-mediated IL-25 release. IL-25 is a tuft cell lineage-defining cytokine[15–17]. IL-25 released from tuft cells triggers the type 2 inflammation via the activation of ILC2s; ILC2 depletion largely abrogates the type 2 immunity and anti-helminthic responses mediated by IL-25[17]. We found that *p53−/−* mice exhibit an impaired ability to release IL-25 in response to infections of *Tm* and *Nb* or succinate treatment. Notably, administering rIL-25 rescues the impaired type 2 immune response in *p53−/−* mice, which indicates that p53 is important for intestinal tuft cells to trigger the type 2 immunity in response to parasitic infections.

We found that the regulation of tuft cell function by p53 is mediated by Lrmp, a previously unrecognized p53 target gene. Analyzing a scRNA expression dataset of mouse intestinal epithelium and employing IF staining assays to co-stain Lrmp in different types of intestinal epithelial cells showed that Lrmp is specially expressed in the intestinal tuft cells. The *Lrmp* gene encodes a type 1 transmembrane protein that localizes to the ER membrane[33]. The biological function of Lrmp is largely unknown, especially its role in type 2 immune response. In this study, we found that Lrmp interacts with IP3Rs, including ITPR2,

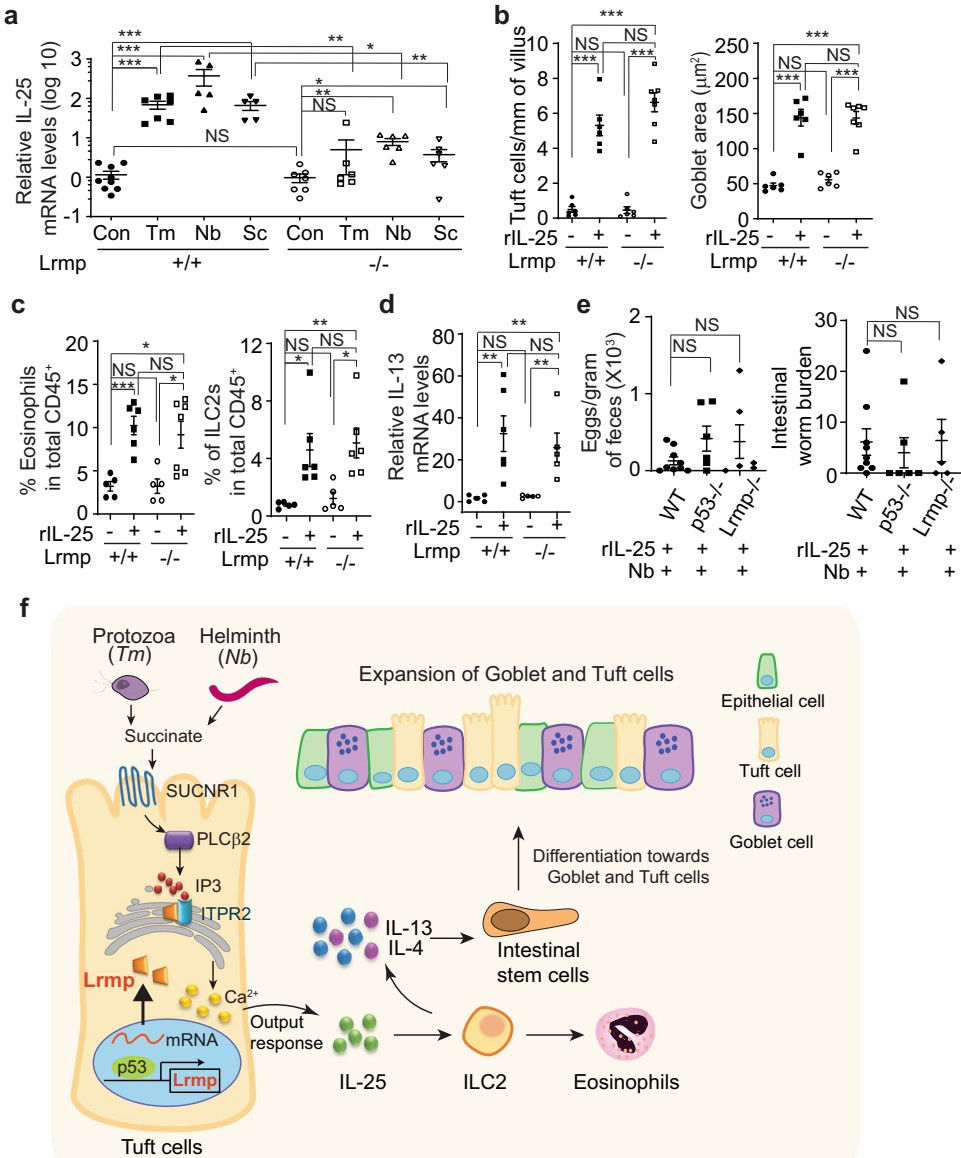

**Fig. 6 rIL-25 administration restores the type 2 immune response in *Lrmp*−/− mice and rescues the impaired ability to expel *Nb* in *p53*−/− and *Lrmp*−/− mice.** **a** Relative IL-25 mRNA levels in the intestinal epithelium of naïve mice, mice infected with *Tm* or *Nb*, and succinate-treated mice. **b**–**d** Administering rIL-25 restored the type 2 immune response in *Lrmp*−/− mice. The expansion of tuft and goblet cells (**b**), the population of eosinophils and ILC2s (**c**), and relative IL-13 mRNA levels (**d**) were examined after administering rIL-25 (0.5 μg/day; *i.p.*) for 7 days in *Lrmp*+/+ and *Lrmp*−/− mice. **e** rIL-25 promoted *Nb* clearance in *p53*−/− and *Lrmp*−/− mice. Mice were infected with *Nb* at day 0 and treated with rIL-25 every 2 days. Egg counts (left) and intestinal worm burden (right) were quantified at 7 d.p.i. **f** Schematic illustration of the role of p53-Lrmp signaling in the intestinal type 2 immunity towards parasitic infections. Data are presented as mean ± SEM. Each dot represents a mouse. $n = 5$–8/group. *$p < 0.05$; **$p < 0.01$; ***$p < 0.001$; NS: non-significant, two-tailed Student's *t*-test.

the major IP3R in the intestinal tuft cells. Further, Lrmp ensures Ca$^{2+}$ flux in cells, which is critical for the tuft cell-IL-25-ILC2 circuit activation in response to parasitic infections. *Lrmp*−/− mice exhibit the impaired type 2 innate immune response towards parasitic infections which can be rescued by rIL-25. We further found that p53 transcriptionally regulates the expression of Lrmp. p53 ensures the high expression of Lrmp in the intestinal tuft cells. It is worth noting that in response to the infections of *Tm* and *Nb* or succinate treatment, p53 protein levels increase in the intestinal tuft cells and p53 selectively regulates the expression of Lrmp but not other classic p53 targets we examined, including p21 and Mdm2. We cannot rule out the possibility that there might be weak upregulation of p21 and MDM2 by p53 in response to *Tm* and *Nb* infections or succinate

treatment that can not be detected by the IF staining. Further investigations in future are needed to understand how p53 is regulated in response to parasitic infections and how p53 selectively regulates the expression of Lrmp in the intestinal tuft cells. Notably, we found that p53 deficiency impairs Ca$^{2+}$ flux in cells, which can be rescued by restoration of Lrmp expression. Further, an impaired intestinal type 2 immune response towards the infections of *Tm* and *Nb* or succinate treatment was observed in *Lrmp*−/− mice. We noticed that compared with *p53*−/− mice, some epithelial phenotypes, especially the expansion of tuft cells, and the production of some cytokines, especially the increase of IL-13, are more severely impaired in *Lrmp*−/− mice in response to the infections of *Tm* and *Nb* or succinate treatment. These differences could be due to that p53 deficiency decreases Lrmp

expression in tuft cells but does not lead to the total loss of Lrmp as in *Lrmp*−/− mice. It is also worth noting that while *Tm* infections in *p53*−/− mice only slightly increase ILC2 levels, IL-13 levels are significantly increased. It is well documented that ILC2s are a robust intestinal source of IL-13. In addition to ILC2s, it has been reported that IL-13 can be secreted by Th2 cells, granulocytes and monocytes/macrophages[47,48]. It is unclear whether the significantly increased levels of IL-13 are secreted by the smaller numbers of ILC2 increased in response to *Tm* infections or by other cell types. Further investigations in the future are needed to identify the source of IL-13 in these mice. Taken together, results in this study suggest that p53 regulates the type 2 immunity, and the transcriptional regulation of Lrmp by p53 in tuft cells to ensure $Ca^{2+}$ flux and IL-25 release from tuft cells is an important underlying mechanism. A previous study reported that Lrmp is expressed in mouse taste tissues, especially in cells that express taste receptors, and interestingly, Lrmp is associated with ITPR3, another IP3R, in these cells[49]. Our finding that Lrmp regulates $Ca^{2+}$ flux raises the possibility that Lrmp also plays an important role in taste signal transduction in taste cells.

Taken together, this study reveals a p53 function in ensuring intestinal type 2 innate immune response, which is important for a rapid and effective immunity against parasitic infections. Given that the type 2 innate immunity also plays critical roles in airway epithelial repair, wound healing, metabolic homeostasis, and allergic inflammation[50], the findings in this study may open potential avenues to investigate the functions of p53 in these important biological processes and diseases.

## Methods

**Mice**. WT and *p53*−/− (B6.129S2-Trp53tm1Tyj/J, stock number 002101) C57BL6/J mice were obtained from The Jackson Laboratory. C57BL/6N-Lrmp$^{tm1a}$ (EUCOMM)Wtsi/WtsiOulu (Lrmp$^{tm1a}$) mice were obtained from EUCOMM program. Lrmp knockout (*Lrmp*−/−) mice were generated by crossing Lrmp$^{tm1a}$ mice containing the "knockout-first" allele with the E2a-Cre mice (B6.FVB-Tg (E2a-cre)C5379Lmgd/J, The Jackson Laboratory, stock number 003724), in which Cre is expressed in the germline, to delete the promoter-driven selection cassette and floxed exon of the tm1a allele, generating a lacZ-tagged knockout allele[35]. Mice were then backcrossed with WT C57BL6/J for at least 5 generations. All mice were maintained in SPF barrier facility at the vivarium of Rutgers University and were confirmed to be protozoa free by IDEXX BioResearch. For ionizing radiation (IR) treatment, mice were subjected to 6 Gy whole-body IR with a 137 Cs γ-source irradiator at a dose rate of 90 cGy/min. Gender-matched and age-matched mice were used for all experiments. Mice were assigned to groups after randomization. All animal procedures were approved by the Institutional Animal Care and Use Committee of Rutgers University.

**Mouse infections and treatments**. For the *Tm* infection, *Tm* was isolated from cecal contents of WT mice naturally colonized with *Tm* as previously described[15]. Mice were infected with *Tm* ($5 \times 10^6$/100 μl PBS) by oral gavage and sacrificed at 21 d.p.i. for analysis. For the *Nb* infection, *Nb* 3$^{rd}$-stage larvae (L3) were raised and maintained as described[51]. Mice were infected subcutaneously (*s.c.*) with 500 *Nb* L3 and sacrificed at 5 d.p.i. for flow cytometric analysis, and at 7 d.p.i. for other assays, respectively. *Nb* worm counts were determined by collecting whole intestinal tissues from the *Nb*-infected mice followed by incubation in warm DPBS solution for 2–6 h to harvest and count worm numbers using a dissection microscope. For *Nb* egg counts, feces were broken up into small pieces using a wooden applicator and suspended in Sheather's floatation solution (454 g sucrose in 355 ml $H_2O$). Eggs were counted using a McMaster's slide. The number of eggs was normalized to the weight of feces. For succinate treatment, mice were provided with sodium succinate hexahydrate (150 mM, Alfa Aesar) *ad libitum* in drinking water for 7 days. For rIL-25 treatment, mice were injected (*i.p.*) daily with 0.5 μg of rIL-25 (R&D) or PBS for 7 days before tissue collection for analysis. For rIL-25 rescue experiments in *Nb*-infected mice, mice infected with *Nb* were injected with rIL-25 (0.5 μg; *i.p.*) once every 2 days. Mice were sacrificed at 7 d.p.i., and *Nb* egg numbers and worm burdens were determined.

**Immunohistochemistry staining and quantification of tuft, goblet cells, and eosinophils**. Intestinal tissues were flushed with PBS and coiled into "Swiss rolls" to make formalin-fixed, paraffin-embedded (FFPE) tissue sections. H&E and IHC staining were performed using standard procedures as previously described[52]. For the tuft cell staining, an anti-Dclk1 (Abcam, ab88484, 1:400 dilution) antibody was used as a primary antibody. The number of tuft cells/mm of villus was calculated

using the Image J software to quantify Dclk1$^+$ cells in each individual villus. Each data point represents the average of at least 30 villi in 5-7 random non-overlapping images from an individual mouse. Goblet cells were identified by Alcian blue staining using Alcian blue solution to stain deparaffinized FFPE tissue sections[21]. The goblet area was quantified using the Image J software. Each data point represents the average area of at least 50 goblet cells from an individual mouse. Eosinophils were visualized by H&E staining and the number of eosinophils was quantified in each field under ×400 magnification.

**Flow cytometric analysis**. The LP from the small intestine and MLNs were collected for flow cytometric analysis. The LP single-cell suspensions were prepared using a Lamina Propria Dissociation Kit (Miltenyi) and a gentleMACS™ Dissociator. Isolated cells were blocked with an anti-CD16/32 antibody and then stained with surface marker antibodies, followed by PI staining for dead cell exclusion. PI$^-$ CD45$^+$ MHC II(I-A/I-E)$^-$ CD11b$^+$ Siglec F$^+$ cell populations were gated as eosinophils. CD45$^+$ Lin$^{-/low}$ CD4$^-$ CD8$^-$ NK1.1$^-$ IL-7Ra$^+$ KLRG1$^+$ IL17Rb$^+$ cell populations were gated as ILC2s. Samples were run on a Beckman-Coulter Cytomics FC500 Flow Cytometer and analyzed by FlowJo 10 software (Tree Star).

**Tuft cell population enrichment**. To enrich tuft cell populations, intestinal single-cell suspensions were prepared as described in the published paper[17]. In brief, the small intestine was opened and rinsed with PBS to remove luminal contents. The small intestine was then incubated in 5 ml PBS containing 2.5 mM EDTA, 0.75 mM DTT, and 10 μg /ml DNaseI (Sigma) at 37 °C for 20 min with rocking. After incubation, tissues were shaken vigorously for 30 s and then incubated in 5 ml HBSS ($Ca^{2+}/Mg^{2+}$ free) containing 1.0 U/ ml Dispase (Gibco) and 10 μg/ml DNaseI at 37 °C for 10 min with rocking. Cells were filtered through a 70 μm filter to get intestinal single-cell suspensions. Intestinal single cells were stained with the EpCAM (Invitrogen, Cat#118202, 1:200 dilution) and SiglecF (BD, Cat#552126, 1:50 dilution) antibodies. EpCAM$^+$ SiglecF$^+$ cell populations were sorted as tuft cell-enriched populations using a BD High-Speed Cell Sorter.

**ELISA assays**. IL-25 levels in the small intestine were determined by ELISA assays using an IL-25 Duoset kit (R&D Systems). Serum IL-4 levels were determined by ELISA assays using IL-4 capture antibody (ebioscience, Cat#14-7041-85) to coat the plate, and biotin-labeled IL-4 antibody (ebioscience, Cat#13-7042-85, 1:1000 dilution) and SA-HRP (BD) for IL-4 detection.

**Small intestinal organoid culture**. The crypt isolation and small intestinal organoid culture were performed as described in the published paper[53]. rIL-13 (10 ng/ml, Biolegend) was added to organoid media for 48 h to stimulate tuft cell expansion.

**Immunofluorescence staining assays**. IF staining of organoids was performed as we described in the published paper[52]. In brief, organoids were fixed with 4% paraformaldehyde, treated with 0.5% TritonX-100, and then blocked with 10% goat serum in IF buffer (0.1% BSA, 0.2% Triton X100, 0.05% Tween-20, and 0.05% NaN$_3$ in PBS) for 2 h. Organoids were incubated with the anti-Dclk1 (Abcam, ab88484, 1:400 dilution) or anti-LRMP (Biorbyt, orb166443, 1:200 dilution) antibodies overnight at room temperature and then incubated with Alexa Fluor® 555 Goat Anti-Mouse IgG (H + L) (Invitrogen, 1:200 dilution) and Alexa Fluor® 488 Goat Anti-Rabbit IgG (H + L) (Invitrogen, 1:200 dilution), respectively. Nuclei were stained with 4′, 6-diamidino-2-phenylindole (DAPI; Vector Labs). To quantify tuft cells in organoids, multiple random non-overlapping pictures of the horizontal cross-section of organoids were acquired using a Nikon A1R-Si Confocal Microscope System. The numbers of Dclk1$^+$ cells were counted using the ImageJ software and then normalized with the surface area of organoids quantified by using the ImageJ software.

IF staining of mouse intestine tissues was performed as previously described[52]. Briefly, tissue sections were deparaffinized in xylene and rehydrated with ethanol. After pre-incubation with 2% BSA and 2% goat serum, tissue sections were incubated with the anti-p53 (Biorbyt, 1:200), anti-p53 (Leica, 1:1000), anti-p21 (Santa Cruz, 1:200), or anti-MDM2 (Santa Cruz, 1:200) antibodies overnight at 4° C, followed by incubating with Alexa Fluor® 555 Goat Anti-Rabbit IgG (H + L) or Alexa Fluor® 555 Goat Anti-mouse IgG (H + L) (1:200). For co-staining, slides were then incubated with the Alexa Fluor® 488 anti-DCLK (Abcam, 1:200) or Alexa Fluor® 488 anti-Lysozyme (Novus, 1:200). Nuclei were stained with 4′, 6-diamidino-2-phenylindole (DAPI; Vector Labs).

**Single-cell data analysis**. Single-cell data analysis was performed by analyzing a publicly available dataset of mouse small intestinal epithelium scRNA expression (GSE92332-AtlasFullLength-TPM)[29].

**Co-immunoprecipitation assays and co-immunoprecipitation followed by chromatography-tandem mass spectrometry assays**. Co-IP assays were performed as we described in the published paper[54]. To determine potential Lrmp-binding proteins, WT MEFs were transfected with control empty vectors, vectors

expressing Lrmp-Flag alone, or together with vectors expressing ITPR2-HA. To identify the potential binding region of Lrmp for ITPR2, human lung H1299 cells that have a high transfection efficiency were co-transfected with vectors expressing different Flag-tagged Lrmp fragments and ITPR2-HA, respectively. The sequences of the primers for Lrmp cloning are listed in Supplementary Table 1. The HA-tagged ITPR2 was synthesized by BIO BASIC Canada Inc., and the ITPR2-HA fragment was subcloned into the pLPCX vector for further experiments. For co-IP assays, the anti-Flag and anti-HA agarose beads were used to pull down Lrmp-Flag and ITPR2-HA proteins, respectively. Uncropped scans of Western-blots presented in the main figures are provided in Source data file. For co-IP followed by LC-MS/MS assays, Lrmp-Flag was pulled down by co-IP using the anti-Flag beads (Sigma) and eluted with 3xFlag peptide (Sigma). Eluted materials were subjected to LC-MS/MS at the Biological MS facility of Rutgers University. Potential Lrmp-interaction proteins were searched against KEGG mapper.

**Cell lines**. Mouse fibroblast Val5 cells contain a temperature-sensitive mutant $p53$ vector (Ala 135 to Val) and express a loss-of-function mutant p53 protein at 37 °C but WT p53 protein at 32 °C. Their parental 10 (1) cells are p53 null. This pair of cell lines are generous gifts from Dr. A. Levine at Institute for Advanced Study. WT, $p53-/-$ and $Lrmp-/-$ MEFs were generated as described previously[55]. Human H1299 cells were obtained from ATCC. Cells were regularly tested for mycoplasma using Lookout Mycoplasma PCR detection kit (MP0035, Sigma).

**Chromatin immunoprecipitation assays**. ChIP assays were performed using an Upstate ChIP assay kit according to the manufacturer's instructions as we previously described[27]. Mouse Val5 and $p53$-null 10(1) cells were cultured at 32 °C for 16 h followed by ChIP assays using the FL393 anti-p53 antibody. The primer sets were designed to encompass the potential p53-binding elements in the intron 1 of the mouse $Lrmp$ gene. The sequences of primers are listed in Supplementary Table 1.

**In situ proximity ligation assays**. The direct interaction of Lrmp and ITPR2 proteins in the small intestine was detected in situ by Duolink™ secondary antibodies and a detection kit (Sigma) according to the manufacturer's instructions. FFPE intestinal tissue sections were subjected to the Lrmp and ITPR2 proximal assays as described in the published paper[34]. In brief, tissue sections were incubated with anti-Lrmp (Biorbyt, orb166443, 1:50 dilution) and anti-ITPR2 (Novus, NB100-2466, 1:50 dilution) antibodies together with Duolink secondary antibodies. Ligation step was performed to ligate the secondary antibodies that are in close proximity. Anti-Dclk1 Alexa Fluor® 488 (Abcam, ab202754, 1:100 dilution) antibodies were used for IF staining on the same tissue slide to detect the tuft cells. DAPI was used for counterstaining the nucleus.

**Construction of reporter vectors and luciferase activity assays**. The fragments containing the putative p53-binding elements in the mouse $Lrmp$ gene were cloned into a pGL2 luciferase reporter vector (Promega). The pGL2 reporter vectors containing one copy of each putative p53-binding element were transfected into $p53$-null MEFs by using Lipofectamine 2000 (Invitrogen) along with pRC-wtp53 (WT p53 expression vector) or empty pRC vectors and pRL-SV40 vectors expressing $Renilla$ luciferase as an internal control to normalize transfection efficiency. The luciferase reporter activity was measured at 24 h after transfection using a Dual-Luciferase® Reporter Assay System (Promega) and normalized with the internal standard. The sequences of primers for fragment amplification are listed in Supplementary Table 1.

**Quantitative real-time PCR assays**. Total RNAs of cells and tissues were extracted using RNeasy kits (QIAGEN) and reverse transcribed to cDNA using Reverse Transcription kits (Invitrogen). Total RNAs of tuft cells were extracted using RNeasy Micro Kits (QIAGEN) and reverse transcribed to cDNA using High-Capacity cDNA Reverse Transcription Kits (Applied Biosystems™). Quantitative real-time PCR was performed in triplicate with Taqman assays (for $Lrmp$, Cat# Mm00493178_m1, and Mm00493168_m1; for actin, Cat# Mm00607939s1) or SYBR green assays (for $Dclk1$, $IL$-$13$, and $IL$-$25$). The sequences of primers used for SYBR green assays are listed in Supplementary Table 1. The mRNA levels of the analyzed genes were normalized with the $\beta$-actin gene.

**Western-blot assays**. Standard Western-blot assays were employed to examine Lrmp protein levels in MEFs and tuft cells using an anti-Lrmp antibody (Biorbyt; Cat#orb166443). Actin levels were examined by an anti-actin antibody (Sigma; Cat#A5441) as a loading control. Uncropped scans of Western-blots presented in the main figures are provided in Source data file.

**Calcium flux Imaging**. MEFs were incubated with 2 μM fluorescent $Ca^{2+}$ indicator Fluo-4 AM (Invitrogen) in the culture medium for 20 min at 37 °C and then maintained in the culture medium for an additional 20 min at 37 °C. Probenecid (2 mM; Sigma) was added into the culture medium throughout the incubation steps to reduce $Ca^{2+}$ leakage. Cells were washed and then maintained in HBSS ($Ca^{2+}$/$Mg^{2+}$-free) throughout imaging. Cells were imaged using a Nikon A1R-Si Confocal

Microscope System with an excitation wavelength of 488 nm and emission wavelength of 525 nm with images captured at an interval of 4 s for 4 min. Ionomycin (2 μM; Thermo Fisher) was added to the reaction buffer at 30 s after imaging started. Fluorescence images were analyzed using the NIS Elements AR 3.2 software.

To determine calcium flux in MEFs with ectopic Lrmp expression, cells were transfected with vectors expressing either FL-Lrmp-Flag or ΔM-Lrmp-Flag using Lipofectamine 2000, and calcium flux was examined at 24 h after transfection.

**Statistical analysis**. The data were expressed as mean ± SD or mean ± SEM as indicated in the figure legends. Source data of mouse experiments are shown in the Source Data file. $p$-values were obtained using two-tailed Student's $t$-tests. Values of $p < 0.05$ were considered to be significant. All data points and "n" values reflect biological replicates.

**Reporting summary**. Further information on research design is available in the Nature Research Reporting Summary linked to this article.

## Data availability
The data supporting the findings of this study are available within the paper and its Supplementary Information files. The source data underlying Figs. 1b–1e, 1g–1j, 2a–2g, 5a–5e, 5–5i, 6a–6d are provided as Source data file. Uncropped blot images are provided as a Source data file. The summary of antibody information, including catalog number, application, and dilution, is provided in Supplementary Table 2. Information of dataset GSE92332 can be found in the following GEO link: https://www.ncbi.nlm.nih.gov/geo/query/acc.cgi?acc=GSE92332. The data that support the findings of this study are available from corresponding authors upon reasonable request. Source data are provided with this paper.

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

## Acknowledgements

W.H. is supported by grants from NIH 1R01CA160558, 1R01CA203965, and DoD W81XWH-18-10238. Z.F. is supported by grants from NIH 1R01CA227912 and 1R01CA214746. C.C. is supported by a New Jersey Commission on Cancer Research (NJCCR) Fellowship Award. This study was also supported by the Flow Cytometry/Cell Sorting shared resource and the Preclinical Imaging shared resource of Rutgers Cancer Institute of New Jersey (supported by NIH P30CA072720). We also thank Dr. Arthur Robert and Dr. Wen Xie for their technical assistance.

## Author contributions

C.C., Y.Z., J.W. carried out the experiments, analyzed data, and wrote the manuscript; J.L., X.Y., H.W., X.Yu., F.Z., J.I., J.P. carried out experiments; L.Z. performed histological analysis; P.X. assisted with ILC2 analysis; M.S. provided *N.b.* and assisted with the experiments of the infections of parasites; Z.F., W.H. developed the concept, designed experiments, analyzed data, and wrote the manuscript.

## Competing interests

The authors declare no competing interests.
