## [Peer Review File · Nature Communications]

Reviewers' Comments:

Reviewer #1:

Remarks to the Author:

In the article entitled "Tumor suppressor p53 regulates intestinal type 2 immunity", the authors investigated the role of the tumour suppressor gene p53 in parasitic infections and the intestinal type 2 immunity. Very little is known about it and the role of p53 in it has never been described. It is a very innovative study as p53 is mostly known for its anti-cancer activity.

The authors clearly and elegantly demonstrate that p53 is crucial for intestinal type-2 immunity in response to the infection of parasites. Knock-out Tp53 mice (deficient in p53) have an impaired intestinal type 2 innate immunity towards the infection of protozoa and helminth parasites. Thus knock-out Tp53 mice cannot eliminate the parasites from their system.

Mechanistically, p53 transcriptionally upregulates LRMP, a novel p53 target gene, to ensure Ca²⁺ flux in the endoplasmic reticulum to regulate the release of IL-25 from intestinal tuft cells in order to recruit and induce the expansion of type 2 innate lymphoid cells (ILC2s) and type 2 helper T cells in the intestinal lamina propria (LP) to secrete type 2 cytokines, including IL-4, IL-5 and IL-13, to drive the type 2 immune response. In turn, IL-13 stimulates the differentiation of intestinal stem cells into tuft cells and goblet cells, which forms a positive feedback loop with tuft cells and promotes the mucus secretion to expel parasites from the intestine.

The data unequivocally support the conclusions. The findings are of interest to the broad scientific community as the findings impact on several wide research fields (infection, immunology, cancer, ageing, cell differentiation, genome programming, ...). This article will pave the way for many future studies.

The manuscript is suitable for publication after minor revisions. It is an exciting and inspiring article.

comments:

In the introduction, the authors should introduce previous works on the role of p53 on Plasmodium, Schistosoma, leishmania,... the authors should also introduce the role of p53 in immunology as it is poorly known although well documented. (for review Nat Rev Immunol. 2016 Dec; 16(12): 741–750.)

Figure2, it looks like the authors have normalised the number of tuft, goblets, eosinophils and ILC2 cells in p53+/+ and p53-/- so that the p53+/+ and p53-/- mice seem to have the same number of tuft, goblets,... cells. However, it may not be the case. Can the authors report the number of the above cells before normalisation in the supplementary documents?

Figure5, it looks like the authors have normalised the number of tuft, goblets, eosinophils and ILC2 cells in LRMP+/+ and LRMP-/- mice so that they seem to have the same number of tuft, goblets ,... cells. However, it may not be the case. Can the authors report the number of the above cells before normalisation in the supplementary documents?

Can the authors provide evidence that the expression of LRMP gene is deficient in LRMP -/- mice?

The figure5 is difficult to read. Can the authors improve legend/annotation? (there are 2 histograms in fig5a or 5b,... can the authors annotate them clearly (tuft and goblet)?

Figure6: it looks like the authors have normalised the numbers. Can the authors report the original numbers before normalisation in the supplementary documents?

Can the authors change fig6f by a graphical abstract such as fig1a with the inclusion of p53 and LRMP?

Reviewer #2:

Remarks to the Author:

In this manuscript, Chang and collaborators address the question of p53 function during intestinal

parasite infections. Using a p53-deficient mouse model, they show that p53 loss of function impairs type 2 immunity in response to helminths or protozoa challenges. A major finding of this study is that p53 regulates the transcription of the *Lrmp* gene within tuft cells. They further show that the *Lrmp* protein interacts with inositol tri phosphate receptor 2 (ITPR2), and that this interaction is required for tuft cells-derived IL-25 release, through a calcium signalling-dependant mechanism. Consequently, they show that mice deficient for *Lrmp* do not mount an efficient type 2 immune response after parasitic challenge.

While the data are generally convincing, and most experiments are well designed and performed, I still have some issues with this manuscript, which may drastically influence the authors conclusions:

Major concerns:

While the p53-dependant *Lrmp* expression within tuft cells makes no doubt to me, data shown in figure 3e show that 8% of Paneth cells express small amount of *Lrmp* transcripts. Thus, is the *Lrmp*-dependant phenotype strictly due to tuft cells, or does it rely on both tuft and Paneth cells? It seems mandatory to me that the authors further characterise the in vivo expression pattern of *Lrmp*, using double IHC for *Lrmp* and either *Dclk1* or Paneth cells markers such as lysozyme. With this experiment, the authors should quantify the rate of co-expression of each marker.

It is somehow difficult for the reader to clearly appreciate how each model (i.e. p53 or *Lrmp* deficiency) impacts the type 2 immune response. Statistical comparisons should be extended to more experimental conditions. First, comparisons between naïve or untreated mice of each genotype would support authors claims (as an example: "Notably, the uninfected p53^{+/+} and p53^{-/-} mice had comparable levels of tuft and goblet cells, eosinophils, ILC2s, and IL-13 in the intestine as well as the serum IL-4 levels"). Also, comparison between naïve control genotype vs. infected or treated p53 or *Lrmp* mice would also be useful to evaluate the phenotype severity. Finally, specifically for figure 6a, statistical analyses between *Lrmp*-deficient mice are also required (is IL-25 release strictly-dependant on *Lrmp* function?)

The discussion part of the manuscript should be reconsidered; the current version mostly summarizes the author's findings. Several points should be discussed in this section, including:

- Comparison of p53 and *Lrmp* deficiency (since p53 deficiency does not totally inhibits *Lrmp* expression, one would expect the *Lrmp* deficiency phenotype to be more drastic than the p53 one, specifically regarding IL-25 release).
- According to the first point, and depending on the author's findings after revision, does *Lrmp* function only rely on tuft cells?
- Are there some explanations regarding mechanisms of regulation of p53 specifically in tuft cells, in naïve or infected conditions?
- Finally, how can the authors explain the weak (but recurrent) epithelial phenotype occurring in their deficient models (exemplified in figure 1b, with tuft cell number increase in p53 deficient mice following Tm infection, in figure 1c and 1e, with goblet cells hyperplasia et IL-13 transcripts quantification, respectively, after the same challenge). The same tendency is also observed for Nb and succinate challenges.

Reviewer #3:

Remarks to the Author:

Chang et al discovered that p53 KO mice are more susceptible to two different parasites, the protozoa *Tritrichomonas muris* (Tm) and the helminth *Nippostrongylus brasiliensis* (Nb). Tuft cells in the intestine are a rare population of cells, that plays an important role in mounting a type 2 innate immune response against parasitic infections. They reveal that p53 is important for tuft cell hyperplasia in response to parasitic infection, that p53 KO mice are less efficient in clearing Nb and that this is due to diminished IL25 secretion from tuft cells. They discover that p53 regulates the expression of *Lrmp*, which they found to be highly expressed in tuft cells. supporting a role for *Lrmp* as a key mediator of p53 control of tuft cell hyperplasia, they produced *Lrmp* KO mice and find that these mice cannot respond to parasites with tuft cell hyperplasia and are more sensitive

to parasitic infection than their WT counterparts.

The study is very well performed with robust data and well-conceived experimental program. While p53 is known to be important in immune responses, its involvement in parasitic infection in general and in regulating tuft cell function in particular has not been reported before to the best of my knowledge. This study is novel in revealing p53 involvement in tuft cell anti-parasite function and reveals a key p53 target gene that is important to mediate this effect. It also provides another example linking taste sensing with parasite sensing – an intriguing concept.

Both parasites and succinate induce LRMP in a p53 dependent manner. Is this a specific p53 transcriptional program in tuft cells such that is selectively activated by p53, or is it part of a classic p53 program in tuft cells. Is p53 stabilized in these cells? Are other p53 genes activated by these signals? Could they test expression of some key p53 target genes in tuft cells? these questions can be addressed with double immunofluorescence for Dclk1 as a marker for tuft cells together with p53 and 2-3 classic targets.

This study identifies Lrmp as a p53 target in tuft cells and verifies this using MEFs. It also shows that LRMP KO phenocopies P53 KO with respect to tuft cell function. This convinces me that at least part of the p53 effect on tuft cells is mediated by transcriptional control of LRMP. However, technically, in order to claim that p53 ensures the function of tuft cells solely through transcription of LRMP they will need to perform rescue experiments as it is possible that there are other p53 targets in this game. I suggest they tone down the claim.

Minor comments:

Does p53 affect the protozoa (Tm) burden? Line 110 claims so, but this is not shown.

Fig. 2d, e - ILC2s are a robust intestinal source of IL-13. While there is no increase in ILC2s in p53 KO mice due to Tm infection, IL13 is significantly increased. What is the source of IL13 in these mice? Are the smaller numbers of ILC2s secreting more IL13, or is another cell taking their place? I don't find this to be a necessary experiment to perform but it could be interesting to know if there are alternative programs that can be recruited in the absence of p53 in either the tuft cells or other cell types.

Fig 2h – add quantification and statistical analysis.

Barenco et al (PMID: 16584535) identified LRMP as a putative p53 target. This strengthens the current report, yet should be cited.

Thank you and all three reviewers for taking time to review our manuscript and providing positive and insightful comments and very constructive suggestions.

We appreciate that all reviewers pointed out that this study reveals a new role of p53 in intestinal type-2 immunity in response to the infection of protozoa and helminth parasites. We have revised our manuscript accordingly and appropriate changes have been incorporated into our manuscript. Following is our response to the comments from reviewers.

Reviewer #1

In the article entitled “Tumor suppressor p53 regulates intestinal type 2 immunity”, the authors investigated the role of the tumour suppressor gene p53 in parasitic infections and the intestinal type 2 immunity. Very little is known about it and the role of p53 in it has never been described. It is a very innovative study as p53 is mostly known for its anti-cancer activity.

The authors clearly and elegantly demonstrate that p53 is crucial for intestinal type-2 immunity in response to the infection of parasites. Knock-out Tp53 mice (deficient in p53) have an impaired intestinal type 2 innate immunity towards the infection of protozoa and helminth parasites. Thus knock-out Tp53 mice cannot eliminate the parasites from their system.

Mechanistically, p53 transcriptionally upregulates LRMP, a novel p53 target gene, to ensure Ca²⁺ flux in the endoplasmic reticulum to regulate the release of IL-25 from intestinal tuft cells in order to recruit and induce the expansion of type 2 innate lymphoid cells (ILC2s) and type 2 helper T cells in the intestinal lamina propria (LP) to secrete type 2 cytokines, including IL-4, IL-5 and IL-13, to drive the type 2 immune response. In turn, IL-13 stimulates the differentiation of intestinal stem cells into tuft cells and goblet cells, which forms a positive feedback loop with tuft cells and promotes the mucus secretion to expel parasites from the intestine.

The data unequivocally support the conclusions. The findings are of interest to the broad scientific community as the findings impact on several wide research fields (infection, immunology, cancer, ageing, cell differentiation, genome programming, ...). This article will pave the way for many future studies. The manuscript is suitable for publication after minor revisions. It is an exciting and inspiring article.

comments:

In the introduction, the authors should introduce previous works on the role of p53 on Plasmodium, Schistosoma, leishmania,... the authors should also introduce the role of p53 in immunology as it is poorly known although well documented.(for review Nat Rev Immunol. 2016 Dec; 16(12): 741–750.)

Response: Thank the reviewer for this very valuable suggestion and pointing out these important previous studies and a good review article on the role of p53 in immunology. We have introduced these previous studies and cited relevant papers including the above-mentioned review article in the Introduction session (Page 3-4, line 47-61).

Figure2, it looks like the authors have normalised the number of tuft, goblets, eosinophils and ILC2 cells

in p53+/+ and p53-/- so that the p53+/+ and p53-/- mice seem to have the same number of tuft, goblets,... cells. However, it may not be the case. Can the authors report the number of the above cells before normalisation in the supplementary documents?

Figure5, it looks like the authors have normalised the number of tuft, goblets, eosinophils and ILC2 cells in LRMP+/+ and LRMP-/- mice so that they seem to have the same number of tuft, goblets ,... cells. However, it may not be the case. Can the authors report the number of the above cells before normalisation in the supplementary documents?

Figure6: it looks like the authors have normalised the numbers. Can the authors report the original numbers before normalisation in the supplementary documents?

Response: The reviewer raised an important point about comparing the numbers of tuft and goblet cells, eosinophils and ILC2 cells between naïve *p53+/+* and *p53-/-* mice (**Fig 2**), and between naïve *Lrmp+/+* and *Lrmp-/-* mice (**Figs 5 & 6**). Data presented in **Figs 2, 5 & 6** did not normalize the numbers of tuft and goblet cells, eosinophils and ILC2 cells in naïve *p53+/+*, *p53-/-*, *Lrmp+/+* and *Lrmp-/-* mice. The reviewer made a very good suggestion to include the source data for these experiments in the supplementary documents. We have provided the Excel file containing the source data for **Figs 2, 5 & 6** in **Extended data table 2** as suggested. **Extended data table 2** also includes the source data for **Fig 1**. We have also performed statistical analysis comparing their numbers between naïve *p53+/+* and *p53-/-* mice as well as naïve *Lrmp+/+* and *Lrmp-/-* mice, and no significant difference was observed. We have included these statistical results in **Figs 1, 2, 5 and 6**.

Can the authors provide evidence that the expression of LRMP gene is deficient in LRMP -/- mice?

Response: This is a very good suggestion to confirm the deficiency of LRMP expression in *Lrmp-/-* mice. We examined the expression of *Lrmp* in WT and *Lrmp-/-* tuft cells by using quantitative real-time PCR. The *Lrmp* mRNA levels can be detected in WT tuft cells but were undetectable in *Lrmp-/-* tuft cells, which confirms the deficiency of *Lrmp* expression in *Lrmp-/-* mice. This result has been included in **Fig 3g**. We have also demonstrated the loss of *Lrmp* at the protein levels in *Lrmp-/-* mice by both Western-blot and IF staining assays. *Lrmp* protein levels were undetectable in *Lrmp-/-* tuft cells as determined by Western-blot analysis (**Fig 3g**) and IF staining of *Lrmp* in *Lrmp-/-* small intestine tissues (**Fig S4f**).

The figure5 is difficult to read. Can the authors improve legend/annotation? (there are 2 histograms in fig5a or 5b,... can the authors annotate them clearly (tuft and goblet)?

Response: Thank the reviewer for this good suggestion. We have annotated each panel in the revised manuscript as suggested.

Can the authors change fig6f by a graphical abstract such as fig1a with the inclusion of p53 and LRMP?

Response: Thank the reviewer for this good suggestion. We have revised **Fig 6f** with the inclusion of the information presented in **Fig 1a** as suggested.

Reviewer #2

In this manuscript, Chang and collaborators address the question of p53 function during intestinal parasite infections. Using a p53-deficient mouse model, they show that p53 loss of function impairs type 2 immunity in response to helminths or protozoa challenges. A major finding of this study is that p53 regulates the transcription of the Lrmp gene within tuft cells. They further show that the Lrmp protein

interacts with inositol tri phosphate receptor 2 (ITPR2), and that this interaction is required for tuft cells-derived IL-25 release, through a calcium signalling-dependant mechanism. Consequently, they show that mice deficient for Lrmp do not mount an efficient type 2 immune response after parasitic challenge.

While the data are generally convincing, and most experiments are well designed and performed, I still have some issues with this manuscript, which may drastically influence the authors conclusions:

Major concerns:

While the p53-dependant Lrmp expression within tuft cells makes no doubt to me, data shown in figure 3e show that 8% of Paneth cells express small amount of Lrmp transcripts. Thus, is the Lrmp-dependant phenotype strictly due to tuft cells, or does it rely on both tuft and Paneth cells? It seems mandatory to me that the authors further characterise the in vivo expression pattern of Lrmp, using double IHC for Lrmp and either Dclk1 or Paneth cells markers such as lysozyme. With this experiment, the authors should quantify the rate of co-expression of each marker.

Response: Thank the reviewer for raising this important point regarding the cell types that express Lrmp, which may in turn contribute to the Lrmp-dependent phenotype. **Fig 3e** shows that ~10% of paneth cells express small amount of Lrmp transcripts by analyzing a scRNA expression dataset of mouse small intestinal epithelium. To address the question whether the Lrmp-dependent phenotype is strictly due to tuft cells or relies on both tuft and paneth cells, we performed double IF staining assays using anti-Lrmp antibodies and either anti-Dclk1 or anti-Lysozyme antibodies (Lysozyme is a marker for paneth cells) in mouse intestinal tissues as suggested. As shown in **Fig S4a**, the expression of Lrmp was observed in 98.3% of tuft cells (294 out of 299 tuft cells), which is consistent with the results in **Fig 3e**. However, no obvious expression of Lrmp was observed in paneth cells with positive staining of lysozyme (**Fig S4a**). These results indicate that even though scRNA analysis showed that ~10% of Paneth cells express small amount of Lrmp mRNA (**Fig 3e**), the protein levels of Lrmp in these paneth cells are much lower than tuft cells. These results indicate that the Lrmp-dependent phenotype is predominantly due to tuft cells.

It is somehow difficult for the reader to clearly appreciate how each model (i.e. p53 or Lrmp deficiency) impacts the type 2 immune response. Statistical comparisons should be extended to more experimental conditions. First, comparisons between naïve or untreated mice of each genotype would support authors claims (as an example: “Notably, the uninfected p53+/+ and p53-/- mice had comparable levels of tuft and goblet cells, eosinophils, ILC2s, and IL-13 in the intestine as well as the serum IL-4 levels”). Also, comparison between naïve control genotype vs. infected or treated p53 or Lrmp mice would also be useful to evaluate the phenotype severity. Finally, specifically for figure 6a, statistical analyses between Lrmp-deficient mice are also required (is IL-25 release strictly-dependant on Lrmp function?)

Response: Thank the reviewer for this very good suggestion. As suggested, we performed following statistical analysis:

1. We compared levels of tuft and goblet cells, eosinophils, ILC2, IL-13 and IL-4 between naïve mice of each genotype (*p53*+/+ vs. *p53*-/-; *Lrmp*+/+ vs. *Lrmp*-/-). Their levels between naïve mice of each genotype show no statistical difference. Statistical results have been added into each panel in **Figs 1, 2, 5, 6**.
2. We compared above-mentioned parameters between naïve wild type mice and infected or treated *p53*-/- or *Lrmp*-/- mice. Statistical results have been added into each panel in **Figs 1, 2, 5, 6**.

3. We compared IL-25 levels between naïve *p53*^{-/-} mice and infected or treated *p53*^{-/-} mice (**Fig 2f**) and between naïve *Lrmp*^{-/-} mice and infected or treated *Lrmp*^{-/-} mice (**Fig 6a**). Statistical results have been added into both panels.

The discussion part of the manuscript should be reconsidered; the current version mostly summarizes the author's findings. Several points should be discussed in this section, including:

- *Comparison of p53 and Lrmp deficiency (since p53 deficiency does not totally inhibit Lrmp expression, one would expect the Lrmp deficiency phenotype to be more drastic than the p53 one, specifically regarding IL-25 release).*

Response: Thanks for this good suggestion. Indeed, we observed a more drastic impairment of type 2 innate immunity, especially the expansion of tuft cells and the induction of IL-13, in response to the infections of *Tm* and *Nb* or succinate treatment in *Lrmp*^{-/-} mice than *p53*^{-/-} mice. As the reviewer pointed out, this difference could be due to that p53 deficiency decreases *Lrmp* expression in tuft cells but does not lead to the total loss of *Lrmp* as in *Lrmp*^{-/-} mice. We have discussed this in the Discussion session (Page 22-23, line 406-412).

- *According to the first point, and depending on the author's findings after revision, does Lrmp function only rely on tuft cells?*

Response: The reviewer raised a very good point. As shown in **Fig S4a**, *Lrmp* expression was detected in tuft cells, but not in paneth cells, which indicates that tuft cells are the major cell type expressing *Lrmp* in the intestinal epithelium and the *Lrmp*-dependent phenotype is predominantly due to tuft cells. We have included this discussion in the Discussion session (Page 21, line 388-390).

- *Are there some explanations regarding mechanisms of regulation of p53 specifically in tuft cells, in naïve or infected conditions?*

Response: This is a very good question. We examined the effect of parasitic infections on p53 protein levels in tuft cells by employing double IF staining assays using antibodies against Dclk1 and p53, respectively. As shown in **Fig S4d**, while p53 protein levels were low and undetectable in tuft cells of naïve mice, infections of *Tm* and *Nb* or succinate treatment increased p53 protein levels in tuft cells, which were detectable by IF staining. It remains unclear how p53 is regulated in tuft cells by parasitic infections. This very interesting question is worth further investigation in future studies. We have included this data in the revised manuscript (**Fig S4d**) and discussed this in the Discussion session (Page 22, line 399-404).

- *Finally, how can the authors explain the weak (but recurrent) epithelial phenotype occurring in their deficient models (exemplified in figure 1b, with tuft cell number increase in p53 deficient mice following Tm infection, in figure 1c and 1e, with goblet cells hyperplasia et IL-13 transcripts quantification, respectively, after the same challenge). The same tendency is also observed for Nb and succinate challenges.*

Response: Thank the reviewer for raising this important point. We observed that the expansion of tuft cells and the induction of IL-13 in response to the infections of *Tm* and *Nb* infections or succinate treatment were impaired but not completely lost in *p53*^{-/-} mice. Results from this study indicated that

the transcriptional regulation of *Lrmp* by p53 is an important mechanism by which p53 regulates the type 2 innate immune response in the intestine. p53 deficiency decreased *Lrmp* expression in tuft cells but did not lead to the total loss of *Lrmp* as in *Lrmp*^{-/-} mice. The weak but recurrent epithelial phenotype observed in *p53*^{-/-} mice could be due to the existence of reduced expression (but not total loss) of *Lrmp* proteins in *p53*^{-/-} tuft cells. We have discussed this in the Discussion session (Page 22-23, line 406-412).

Reviewer #3

Chang et al discovered that p53 KO mice are more susceptible to two different parasites, the protozoa Tritrichomonas muris (Tm) and the helminth Nippostrongylus brasiliensis (Nb). Tuft cells in the intestine are a rare population of cells, that plays an important role in mounting a type 2 innate immune response against parasitic infections. They reveal that p53 is important for tuft cell hyperplasia in response to parasitic infection, that p53 KO mice are less efficient in clearing Nb and that this is due to diminished IL25 secretion from tuft cells. They discover that p53 regulates the expression of Lrmp, which they found to be highly expressed in tuft cells. supporting a role for Lrmp as a key mediator of p53 control of tuft cell hyperplasia, they produced Lrmp KO mice and find that these mice cannot respond to parasites with tuft cell hyperplasia and are more sensitive to parasitic infection than their WT counterparts.

The study is very well performed with robust data and well-conceived experimental program. While p53 is known to be important in immune responses, its involvement in parasitic infection in general and in regulating tuft cell function in particular has not been reported before to the best of my knowledge. This study is novel in revealing p53 involvement in tuft cell anti-parasite function and reveals a key p53 target gene that is important to mediate this effect. It also provides another example linking taste sensing with parasite sensing – an intriguing concept.

Both parasites and succinate induce LRMP in a p53 dependent manner. Is this a specific p53 transcriptional program in tuft cells such that is selectively activated by p53, or is it part of a classic p53 program in tuft cells. Is p53 stabilized in these cells? Are other p53 genes activated by these signals? Could they test expression of some key p53 target genes in tuft cells? these questions can be addressed with double immunofluorescence for Dclk1 as a marker for tuft cells together with p53 and 2-3 classic targets.

Response: Thank the reviewer for raising this very important question regarding the p53 transcriptional program in tuft cells in response to parasitic infections. We also want to thank the reviewer for the excellent suggestion to address this question. As suggested, we examined whether parasitic infections stabilize p53 protein in tuft cells by employing double IF staining assays using antibodies against Dclk1 and p53, respectively. As shown in **Fig S4d**, p53 protein levels were low and undetectable in tuft cells of naïve mice, infections of *Tm* and *Nb* as well as succinate treatment increased p53 protein levels in tuft cells. We also examined the levels of p21 and Mdm2, two classic p53 targets, in tuft cells by employing double IF staining assays. As shown in **Fig S4e**, while both of p53 targets were induced in the intestinal epithelial cells by γ -irradiation, no obvious induction of p21 and Mdm2 was observed in tuft cells after infections of *Tm* and *Nb* or succinate treatment as determined by IF staining assays. These results indicate that p53 selectively upregulates *Lrmp* but not p21 or Mdm2 in tuft cells in response to parasitic infections.

This study identifies Lrmp as a p53 target in tuft cells and verifies this using MEFs. It also shows that LRMP KO phenocopies P53 KO with respect to tuft cell function. This convinces me that at least part of

the p53 effect on tuft cells is mediated by transcriptional control of LRMP. However, technically, in order to claim that p53 ensures the function of tuft cells solely through transcription of LRMP they will need to perform rescue experiments as it is possible that there are other p53 targets in this game. I suggest they tone down the claim.

Response: Thanks for this suggestion. We agree with the Reviewer. While our results support that the transcriptional regulation of *Lrmp* by p53 contributes to the effect of p53 on tuft cells, without the rescuing experiments, we can not conclude that p53 ensures the function of tuft cells solely through transcription of *Lrmp*. As the reviewer suggested, we have toned down the claim in the manuscript, including in the Abstract, the Introduction, the Results and the Discussion parts (Pages 2, 5, 18, 21, 23).

Minor comments:

Does p53 affect the protozoa (Tm) burden? Line 110 claims so, but this is not shown.

Response: Thanks for raising this good question. We compared the *Tm* burden in *p53*^{+/+} and *p53*^{-/-} mice by measuring the amount of *Tm* in the cecum at 21 d.p.i.. As shown in **Fig 1f**, *Tm* amount in the cecum was significantly higher in *p53*^{-/-} mice than in *p53*^{+/+} mice, which suggests that p53 protects mice from the infection of protozoa (*Tm*).

Fig. 2d, e - ILC2s are a robust intestinal source of IL-13. While there is no increase in ILC2s in p53 KO mice due to Tm infection, IL13 is significantly increased. What is the source of IL13 in these mice? Are the smaller numbers of ILC2s secreting more IL13, or is another cell taking their place? I don't find this to be a necessary experiment to perform but it could be interesting to know if there are alternative programs that can be recruited in the absence of p53 in either the tuft cells or other cell types.

Response: Thanks for raising this very good point. As shown in **Fig 1d & 1e**, while *Tm* infections in *p53*^{-/-} mice only slightly increased ILC2 levels which was not statistically significant compared with naïve *p53*^{-/-} mice, IL-13 levels were significantly increased in *p53*^{-/-} mice upon *Tm* infections. It is well-documented that ILC2s are a robust intestinal source of IL-13. It is unclear the source of IL-13 in these mice. In addition to ILC2s, it has been reported that IL-13 can be secreted by Th2 cells, granulocytes and monocytes/macrophages (Bao & Reinhardt, *Cytokine*, 75:25; McCormick & Heller, *Cytokine*, 75:38). We totally agree with the reviewer that this is a very interesting question, which is worth further investigation in future. In particular, it will be interesting to investigate whether the significantly increased levels of IL-13 are secreted by the smaller numbers of ILC2 increased in response to *Tm* infections or by other cell types. We have discussed this important point in the Discussion part (Page 23, line 412-419).

Fig 2h – add quantification and statistical analysis.

Response: Thanks for this good suggestion. We have quantified tuft cell numbers in **Fig 2h** and performed statistical analysis as suggested.

Barenco et al (PMID: 16584535) identified LRMP as a putative p53 target. This strengthens the current report, yet should be cited.

Response: Thanks for pointing out this literature which identified *Lrmp* as a putative p53 targets. We have cited this paper in Page 10-11, line 176-178.

Again, we want to thank reviewers for their great efforts and very insightful and constructive comments to improve our manuscript. We hope that with these changes and response, our manuscript would be acceptable for publication. Thank you very much!

Sincerely yours,

Wenwei Hu, Ph.D.
Associate Professor
Rutgers Cancer Institute of New Jersey
New Brunswick, NJ 08903
Email: wh221@cinj.rutgers.edu
Phone: 732-235-6169

Reviewers' Comments:

Reviewer #1:

Remarks to the Author:

The authors have addressed all reviewers' comments. The conclusions are supported by the data. the manuscript is suitable for publication

Reviewer #2:

Remarks to the Author:

The authors answered adequately to all my questions and comments. The manuscript has been greatly improved, and I can recommend it for publication.

Reviewer #3:

Remarks to the Author:

The revised manuscript is strengthened with the new data and I feel it could open a new direction in p53 research, in relation to its expanding immune functions.

Because of the importance of this manuscript, I still want to point a few minor issues which I urge the authors to consider:

In fig. S4D – they should add p53 staining in irradiated p53 KO mice (i.e. a fourth line in the topmost 3X3 matrix of images). This may sound like a trivial experiment; however, p53 immunostaining is notoriously prone for artifacts and this will rule out the possibility that the p53 immunostaining they're observing in the tuft cells is an artifact. Would be even better if they stain protozoa infected p53 KO mice.

There is mislabeling in fig. S4E (Dclk1 and DAPI are swapped in the p21 panel).

It is puzzling that p53 upregulates the expression of LRMP, but not that of classic p53 target genes such as MDM2 and p21. After all, despite its elaborate network, p53 is just a protein, that when stabilized, binds DNA and activates transcription. I wonder whether this could be a dose effect (i.e. that it is upregulating them, but to a smaller extent than IR, which is undetectable by their staining). The importance of this finding is noted by the authors and is related to in the discussion section (lines 405 – 410). Admittedly, this makes biological sense – upregulating the classical p53 response would be detrimental. Still, it would be good to look again at these IF images, to make sure there is no weak upregulation of p21.

Thank you and all three reviewers for taking time to review our manuscript. We are glad that all three reviewers are happy with the changes we made in the revised manuscript. While two reviewers are fully satisfied with the revised manuscript, one reviewer raised very valuable comments to help us further improve and strengthen the manuscript.

We have revised our manuscript accordingly and appropriate changes have been incorporated into our manuscript. Following is our response to the comments from reviewers.

Reviewer #1:

The authors have addressed all reviewers' comments. The conclusions are supported by the data. The manuscript is suitable for publication.

Reviewer #2:

The authors answered adequately to all my questions and comments. The manuscript has been greatly improved, and I can recommend it for publication.

Thank both reviewers for positive comments.

Reviewer #3:

The revised manuscript is strengthened with the new data and I feel it could open a new direction in p53 research, in relation to its expanding immune functions.

Because of the importance of this manuscript, I still want to point a few minor issues which I urge the authors to consider:

In fig. S4D – they should add p53 staining in irradiated p53 KO mice (i.e. a fourth line in the topmost 3X3 matrix of images). This may sound like a trivial experiment; however, p53 immunostaining is notoriously prone for artifacts and this will rule out the possibility that the p53 immunostaining they're observing in the tuft cells is an artifact. Would be even better if they stain protozoa infected p53 KO mice.

Response: Thanks for this very good suggestion. As suggested, we performed p53 staining in the intestine tissues of irradiated p53 KO mice as well as Tm or Nb-infected p53 KO mice. No obvious p53 staining was observed in all of these samples, which ruled out the possibility that the p53 IF staining signals we observed were an artifact. These data have been added to **Fig S4d**.

There is mislabeling in fig. S4E (Dclk1 and DAPI are swapped in the p21 panel).

Response: Thank the reviewer for pointing out this mistake. We have corrected the mislabeling for panels Dclk1 and DAPI in **Fig S4e**.

It is puzzling that p53 upregulates the expression of LRMP, but not that of classic p53 target genes such as MDM2 and p21. After all, despite its elaborate network, p53 is just a protein, that when stabilized, binds DNA and activates transcription. I wonder whether this could be a dose effect (i.e. that it is upregulating them, but to a smaller extent than IR, which is undetectable by their staining). The importance of this finding is noted by the authors and is related to in the discussion section (lines 405 – 410). Admittedly, this makes biological sense – upregulating the classical p53 response would be detrimental. Still, it would be good to look again at these IF images, to make sure there is no weak upregulation of p21.

Response: Reviewer raised an important point that no obvious induction of p21 and MDM2 by p53 in tuft cells in response to parasitic infections could be due to a dose effect and the detection limitation of IF staining. In another word, p53 induces p21 and MDM2 in tuft cells in response to parasitic infections to a smaller extent than IR, which is below the detection limitation of the IF staining. As suggested, we revisited our IF staining results. We have checked p21 and MDM2 staining in over 100 tuft cells from Tm or Nb-infected or Sc-treated mice, and no obvious staining/weak upregulation of p21 or MDM2 was observed. We clarified in Results (line 220-221) that no obvious increase of the levels of p21 and MDM2 was observed in the tuft cells from mice with Tm and Nb infections or succinate treatment as determined by the IF staining. We have also discussed the possibility that there might be weak upregulation of p21 and MDM2 by p53 in response to Tm and Nb infections or succinate treatment that can not be detected by the IF staining (line 400-402).

Again, we want to thank reviewers for their great efforts and very insightful and constructive comments to improve our manuscript. We hope that with these changes and response, our manuscript would be acceptable for publication. Thank you very much!

Sincerely yours,

Wenwei Hu, Ph.D.
Associate Professor
Rutgers Cancer Institute of New Jersey
New Brunswick, NJ 08903
Email: wh221@cinj.rutgers.edu
Phone: 732-235-6169